# Quantifying cerebral contributions to pain beyond nociception

Choong-Wan Woo[1,2,†], Liane Schmidt[3,4,*], Anjali Krishnan[5,*], Marieke Jepma[6,7], Mathieu Roy[8], Martin A. Lindquist[9], Lauren Y. Atlas[10,11] & Tor D. Wager[1,2]

Cerebral processes contribute to pain beyond the level of nociceptive input and mediate psychological and behavioural influences. However, cerebral contributions beyond nociception are not yet well characterized, leading to a predominant focus on nociception when studying pain and developing interventions. Here we use functional magnetic resonance imaging combined with machine learning to develop a multivariate pattern signature—termed the stimulus intensity independent pain signature-1 (SIIPS1)—that predicts pain above and beyond nociceptive input in four training data sets (Studies 1–4, $N = 137$). The SIIPS1 includes patterns of activity in nucleus accumbens, lateral prefrontal and parahippocampal cortices, and other regions. In cross-validated analyses of Studies 1–4 and in two independent test data sets (Studies 5–6, $N = 46$), SIIPS1 responses explain variation in trial-by-trial pain ratings not captured by a previous fMRI-based marker for nociceptive pain. In addition, SIIPS1 responses mediate the pain-modulating effects of three psychological manipulations of expectations and perceived control. The SIIPS1 provides an extensible characterization of cerebral contributions to pain and specific brain targets for interventions.

[1] Department of Psychology and Neuroscience, University of Colorado, Boulder, Colorado 80309, USA. [2] Institute of Cognitive Science, University of Colorado, Boulder, Colorado 80309, USA. [3] INSEAD, Fontainebleau 77300, France. [4] Cognitive Neuroscience Laboratory, INSERM U960, Department of Cognitive Sciences, Ecole Normale Supérieure, Paris 75005, France. [5] Department of Psychology, Brooklyn College of the City University of New York, Brooklyn, New York 11210, USA. [6] Cognitive Psychology Unit, Institute of Psychology, Leiden University, Leiden 2300, The Netherlands. [7] Leiden Institute for Brain and Cognition, Leiden University, Leiden 2300, The Netherlands. [8] Department of Psychology, McGill University, Montréal, Quebec H3A 0G4, Canada. [9] Department of Biostatistics, Johns Hopkins University, Baltimore, Maryland 21211, USA. [10] National Center for Complementary and Integrative Health, National Institutes of Health, Bethesda, Maryland 20892, USA. [11] National Institute on Drug Abuse, National Institutes of Health, Rockville, Maryland 20852, USA. † Present addresses: Center for Neuroscience Imaging Research, Institute for Basic Science, Suwon 16419, Republic of Korea; Department of Biomedical Engineering, Sungkyunkwan University, Suwon 16419, Republic of Korea. * These authors contributed equally to this work. Correspondence and requests for materials should be addressed to T.D.W. (email: tor.wager@colorado.edu).

Pain is widely thought to emerge from distributed brain networks whose inputs include sensory, affective and evaluative processes[1]. Although Melzack's[1] influential 'body-self neuromatrix' framework for pain emphasized many processes beyond nociception—including expectancy, attention, anxiety, and personality—the 'neuromatrix' came to be increasingly identified with a set of regions that encode the intensity of nociceptive input[2]. However, there has been sustained interest in the roles of other brain regions that have been commonly considered to be non-nociceptive, including the dorsolateral prefrontal cortex (dlPFC)[3,4], hippocampus[5,6], ventromedial prefrontal cortex (vmPFC)[7–9], nucleus accumbens (NAc)[10–12]. These regions are often thought to play support roles, influencing pain by modulating activity in nociceptive circuits[10,13], but they may also play a central role in pain construction independent of nociceptive circuits. Several recent studies of chronic pain in animal models suggest that this is the case[14–16] and implicate the vmPFC, NAc and other regions in mediating pain-related behaviours independent of classic probes of nociceptive pain[17,18]. Chronic pain appears to involve a shift away from classic nociceptive regions and towards a type of pain directly maintained in frontal-limbic networks[8,18–20], and new theories describe pain as an emergent phenomenon related to activity in large-scale networks that include non-nociceptive regions[21,22]. It is therefore vital to gain an increasingly precise understanding of the roles of non-nociceptive brain regions in human pain.

Human neuroimaging approaches to understanding pain have been hampered by two important, but addressable, limitations. First, they have typically not specified hypotheses with sufficient precision, limiting direct replications. For example, although the dlPFC has been implicated in pain, findings vary widely in their location and topography from study to study. Results are aggregated in meta-analyses[23,24], but there is no consensus on how close findings should be to be considered 'replications'. Regions of interest used in a priori analyses are typically large, encompassing heterogeneous groups of neurons with different functions without specifying any particular topography, which dilutes signal and reduces their functional specificity. By contrast, multivariate pattern signatures can specify a precise set of voxels and the topography of the relative expected activity levels across voxels, providing a basis for exact replication. Analysing pattern information is analogous to analysing neural population codes[25] and a number of studies show convincingly that they can capture fine-grained functional organization (for example, ocular dominance columns[26]) and can more accurately predict perceptions and behaviours than standard brain maps[27,28]. Second, it is now clear that functional magnetic resonance imaging (fMRI) responses in imprecisely defined gross anatomical regions (for example, the anterior cingulate) are not specific to pain, but that precisely defined multivariate patterns can have much greater sensitivity and specificity[28,29].

For these reasons, a number of groups have turned to multivariate pattern analysis to identify precisely defined patterns that predict pain intensity[30–33]. One recent example is the Neurologic Pain Signature (NPS)[32], a multivariate pattern whose weights, which specify relative activity levels, are optimized to be maximally predictive of pain based on fMRI signal. The NPS is precisely specified so that it can be applied to new data from 7 new individuals by taking a weighted average over a test brain image (the NPS supplies the weights), yielding a single predicted pain value. This feature permits detailed characterization of its measurement properties. The NPS accurately predicts pain experience in response to noxious thermal[32,34], mechanical[35] and electrical stimuli[35,36], but does not respond to non-noxious warm stimuli[32], threat cues[32,35,36], social rejection-related stimuli[32], observed pain[35], or aversive images[37]. However, similar to other pain-predictive patterns, the NPS was developed to predict pain experience driven largely, although not entirely, by noxious stimuli based on fMRI activity mostly within noxious stimulus intensity-encoding regions. It reflects only a subset of the various brain processes that contribute to pain and does not explain much of the variation in pain experience that is found even when the stimulus intensity is held constant (for an example case, see Fig. 1a). In addition, recent studies have shown that the NPS does not explain the pain-modulating effects of several psychological interventions, including placebo treatment[32], cognitive self-regulation[17] and perceived control[34].

Combining the precision of multivariate pattern approaches with the study of regions outside classic nociceptive pain-related brain regions could help provide a more precise understanding of the roles of the vmPFC, NAc, dlPFC, hippocampus and others in pain processing in humans. In addition, if it is possible to identify pain-predictive patterns that are independent of noxious stimulus intensity and nociceptive brain targets, this could point to a direct role for these regions in constructing pain rather than simply modulating ongoing pain. Thus, in this study, we asked: (1) can we identify a multivariate pattern of fMRI activity that predicts pain experience after removing the effects of noxious stimulus intensity and the NPS (Fig. 1b)? (2) If so, which brain regions are involved? (3) Does a model that includes independent contributions from non-nociceptive brain regions predict pain better than using classic noxious stimulus-encoding regions alone? Furthermore, (4) does a model that includes stimulus-independent brain regions better explain the effects of psychological interventions on pain, including expectancy and perceived control?

To address these questions in a way that is replicable and generalizable beyond a single study, we combined a mega-analytic approach with machine learning techniques. Our data set included ~11,000 single-trial images of fMRI activity associated with multiple levels of noxious heat and pain ratings, across 183 participants from 6 independent studies. We first developed a new multivariate fMRI signature, termed the stimulus intensity independent pain signature-1 (SIIPS1), which is predictive of variation in pain above and beyond noxious stimulus intensity (for example, heat temperature) and nociceptive brain processes estimated by the NPS, using Studies 1–4 ($N = 137$; Supplementary Table 1) as training data. We named this signature because our approach relies on precisely specifying patterns and testing them across studies, and having a name is essential to communicate that it is this precise pattern that can be used in future studies (for example, see ref. 38). We evaluated the performance of the SIIPS1 in cross-validated analyses of Studies 1–4 and in two independent test data sets (Studies 5–6, $N = 46$) using it to answer the four questions above. Results show that the SIIPS1 explains a substantial amount of the variation in trial-by-trial pain ratings not captured by the NPS. The SIIPS1 was a significant and consistent mediator of the effects of psychological interventions, including manipulations of expectancy and perceived control, whereas the NPS was not. Overall, the current study provides a viable new signature that can quantify cerebral contributions to pain beyond nociception.

## Results

**Signature development**. To develop the SIIPS1 pattern, we employed a multi-level approach (for details, see Supplementary Fig. 1). We began with single-trial estimates of brain responses during individual epochs of noxious heat from 137 participants in Studies 1–4 (6,740 images total, ~50 trial-level images per person on average). First, we removed the effects of stimulus intensity

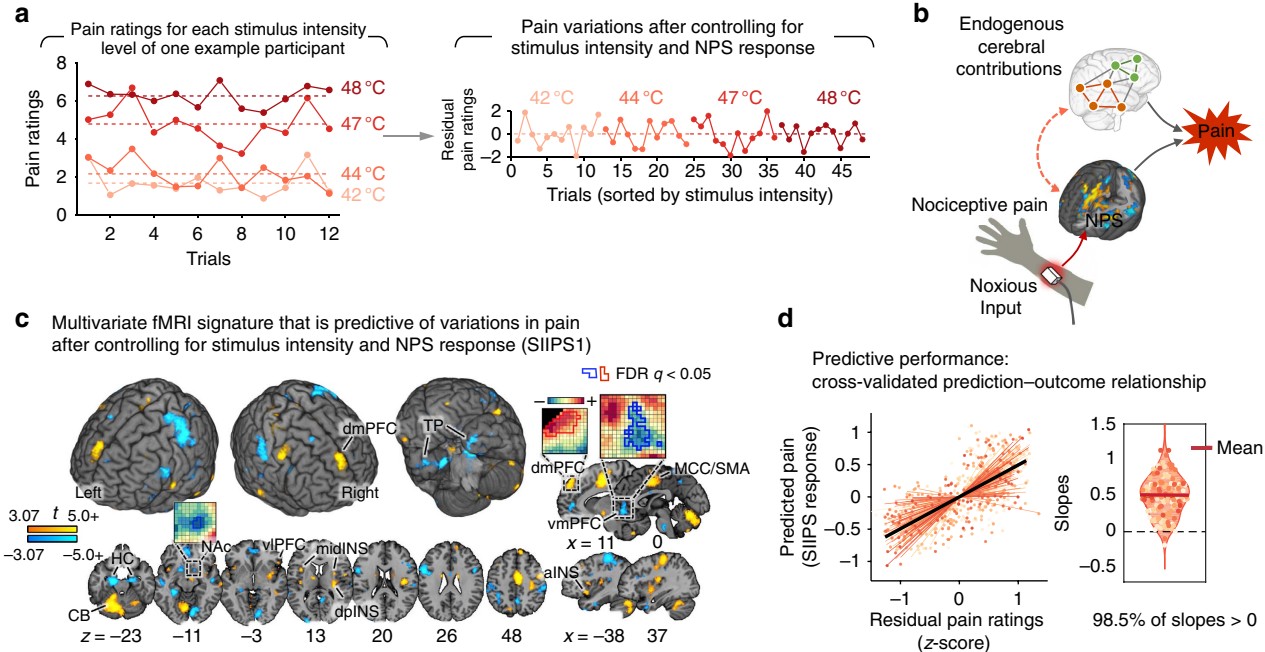

**Figure 1 | Identifying cerebral contributions to pain beyond nociception. (a)** The left panel provides an example of pain ratings for different levels of noxious stimuli and the right panel shows that there still remain large variation in pain ratings even after controlling for noxious stimulus intensity and the neurologic pain signature (NPS) response[32]. **(b)** The main goal of the current study is to develop a multivariate model of endogenous cerebral contributions to pain beyond nociception. Some of the cerebral contributions may interact with nociceptive brain systems (red nodes), whereas others contribute to pain independent of nociceptive processing (green nodes). **(c)** The multivariate pattern of fMRI activity predictive of residual pain ratings after removing the effects of the stimulus intensity and NPS response, termed the stimulus intensity independent pain signature-1 (SIIPS1). The map shows thresholded voxel weights (at $q < 0.05$ false discovery rate (FDR), equivalent to uncorrected voxel-wise $P < 0.0025$) based on weighted $t$-tests across maps for 137 subjects in the training data sets (Studies 1–4). Thresholding was performed for display only; all weights were used in the subsequent analyses. Some examples of unthresholded patterns are presented in the insets; small squares indicate individual voxel weight. aINS, anterior insula; CB, cerebellum; dmPFC, dorsomedial PFC; dpINS, dorsal posterior insula; HC, hippocampus; MCC, mid-cingulate cortex; midINS, middle insula; NAc, nucleus accumbens; SMA, supplementary motor area; TP, temporal pole; vmPFC, ventromedial PFC; vlPFC, ventrolateral PFC. **(d)** Z-scored quartile residual pain ratings versus cross-validated (leave-one-participant-out) prediction (also z-scored and quartile binned) with the SIIPS1. Each coloured line represents a fitted line for each individual. The violin plot in the right panel shows the distribution of the slopes from regression analyses for the prediction–outcome relationship. All participants except for two individuals (98.5%) showed positive prediction–outcome relationships. Each coloured dot represents an individual's slope.

and the NPS response (a proxy for already modelled nociceptive neural processes) from each participant's single trial-level brain images using a set of regressors modeling all possible differences among intensities (for details, see Methods). Second, we used principal component regression (PCR) to estimate a multivariate fMRI pattern that predicted residual pain ratings for each individual; this method works well with high-dimensional, multicollinear predictors[39]. Ten-fold cross-validation was used to estimate each individual's prediction–outcome correlation. Third, we constructed a population-level pattern map using a weighted average of the predictive maps for the 137 participants using prediction–outcome correlations as a weight (all prediction–outcome correlations were positive). Weighted $t$-tests identified which brain areas made consistent contributions to prediction across participants and studies, treating participant as a random effect.

As shown in Fig. 1c and Supplementary Fig. 2, the resulting signature pattern was consistent in many brain areas across participants and studies, indicating that there are brain systems for cerebral contributions to pain beyond nociception that are conserved across individuals. These regions fell into three classes, based on their relationships with pain and noxious stimulus intensity.

The first class of regions included established targets of nociceptive afferents, such as the insula, cingulate cortex

and thalamus, and overlapped spatially with regions included in the NPS. These regions showed positive weights in the SIIPS1, indicating that their activation predicted increased pain even when the noxious stimulus intensity is constant and NPS responses are controlled for. Further analyses showed that brain activity in these regions was indeed correlated with noxious input intensity (Fig. 2a,b and Supplementary Table 4); thus, these regions are not truly 'nociception independent', even though we regressed out stimulus intensity and the NPS response from the training data. This finding is sensible if endogenous variation in these nociceptive regions contributes to pain experience beyond simply encoding input intensity[40,41] or the NPS is an imperfect proxy for nociception-induced pain. We note that local pattern similarity analyses showed that the SIIPS1 and NPS weight patterns within these regions are not correlated (Supplementary Fig. 3), indicating that the SIIPS1 is capturing pain-related brain activity that the NPS does not capture, even within the overlapping brain regions.

The second class of regions also showed positive pain-predictive weights, but are not known to be targets of spinal nociceptive afferents, suggesting that they are likely to make extra-nociceptive contributions to pain. These included dorsomedial prefrontal cortex (dmPFC), middle temporal gyrus, caudate and ventrolateral PFC. These regions showed minimal correlations with noxious stimulus intensity (Fig. 2 and Supplementary

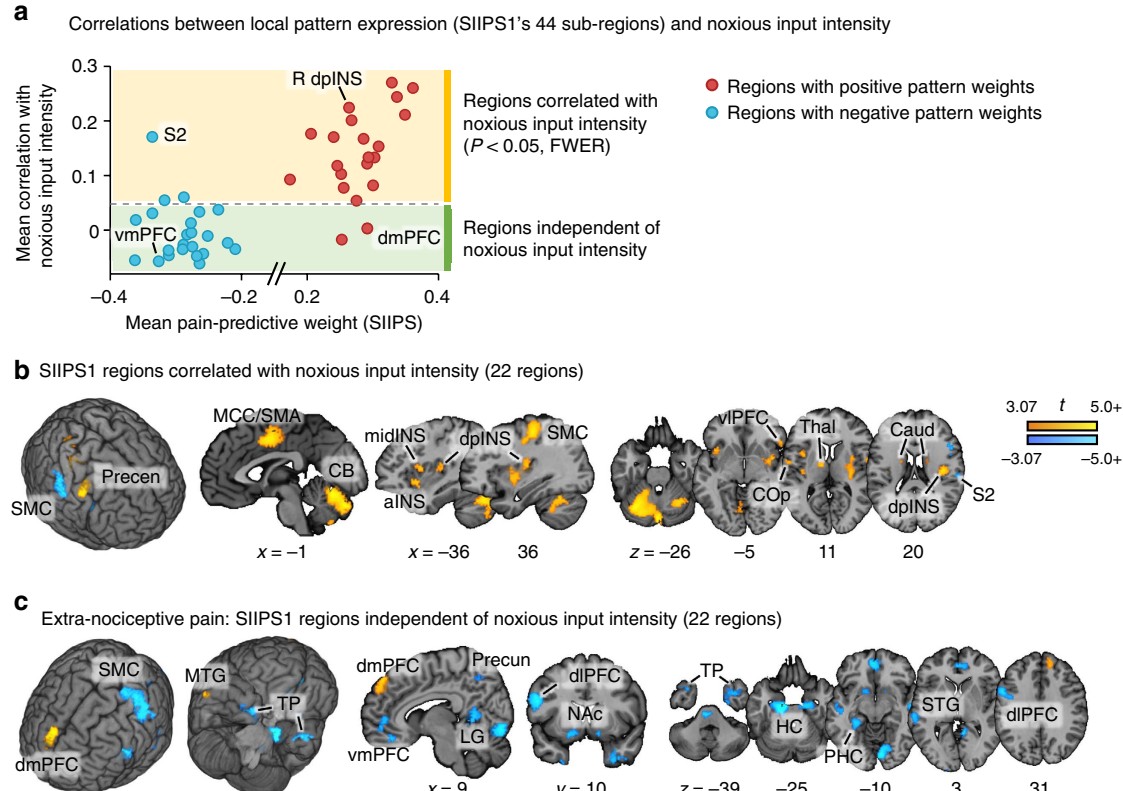

**Figure 2 | Deconstructing the SIIPS1. (a)** Each dot of the scatter plots represents a contiguous brain region from the SIIPS1 thresholded at $q < 0.05$, FDR corrected (see Fig. 1c). Red dots represent regions with positive predictive weights, and blue dots represent regions with negative predictive weights. The y axis of the scatter plots shows the mean correlations between the local pattern expression (with absolute pattern weights) and trial-by-trial noxious stimulus intensity across 183 participants from Studies 1–6. The x axis of the scatter plots shows the mean pattern weights of contiguous regions. Dashed gray lines indicate one-sample t-test results that are corrected for multiple comparisons using family-wise error rate $< 0.05$ (Bonferroni methods; equivalent to uncorrected $P < 0.0011$). Therefore, dots above the dashed lines indicate regions significantly correlated with noxious input intensity (temperature) and dots below the dashed line indicate regions independent of noxious input intensity. Brain region maps for **(b)** regions that showed significant non-zero correlations with noxious input intensity and for **(c)** regions that showed no correlations with noxious input intensity, but still contributed to the prediction of single-trial level pain ratings. Region labels, mean weight values and mean correlation values with noxious stimulus intensity (and their t- and p-values) can be found in Supplementary Table 4. aINS, anterior insula; Caud, caudate; CB, cerebellum; dmPFC, dorsomedial PFC; dlPFC, dorso-lateral PFC; dpINS, dorsal posterior insula; HC, hippocampal area; LG, lingual gyrus; MCC, middle cingulate cortex; midINS, middle insula; MTG, middle temporal gyrus; NAc, nucleus accumbens; PHC, parahippocampal area; Precen, precentral cortex; Precun, precuneus; S2, secondary somatosensory cortex; SMA, supplementary motor area; SMC, sensory motor cortex; STG, superior temporal gyrus; Thal, thalamus; TP, temporal pole; vlPFC, ventrolateral PFC; vmPFC, ventro-medial PFC.

Table 4): the caudate and ventrolateral PFC showed significant, but low, correlations with stimulus intensity ($r = 0.08$, $P < 0.001$ and $r = 0.13$, $P < 0.001$, respectively, one sample t-test on within-subject correlations after Fisher's r-to-z transformation, $N = 183$). Other regions including dmPFC and middle temporal gyrus showed no relationship with noxious input intensity ($r = 0.003$, $P = 0.77$ and $r = -0.017$, $P = 0.18$, respectively, one sample t-test, $N = 183$). Such regions could be involved in constructing value and motivation related to pain or in mediating internal thought processes that increase pain independent of nociception.

The third class of regions had negative predictive weights, indicating that increased brain activity was associated with decreased pain. Such regions included vmPFC, NAc, parahippocampal cortex, posterior dlPFC and others. Most of these regions were uncorrelated with noxious input intensity (Fig. 2a,c and Supplementary Table 4), suggesting that these regions make extra-nociceptive contributions to pain. Growing evidence suggests that these regions contribute to cognitive, evaluative or motivational aspects of pain instead of sensory ones[14,15,42] and that they play critical roles in chronic pain[4,43].

The signature pattern we identified here can be prospectively applied to individual trial images or other images (for example, condition averages) to make quantitative predictions about pain in out-of-sample individual participants. In the current training data, the cross-validated SIIPS1 response (deriving pattern maps from training data, except for one out-of-sample participant, and calculating the signature response for the out-of-sample participant) predicted residual pain ratings with mean $r = 0.68$ when grouping trials into quartiles based on residual pain ratings (Fig. 1d).

**Characterization of local pattern topography.** The SIIPS1 also revealed finer-grained structure captured in local pattern weights within anatomical regions (Fig. 3). The relatively large sample size combined with multivariate methods here affords increased reliability of these pattern weights, which can reveal structure not often apparent in smaller samples or univariate approaches. In particular, the SIIPS1 includes a region possibly corresponding to the NAc shell that predicted increased pain, whereas a region that

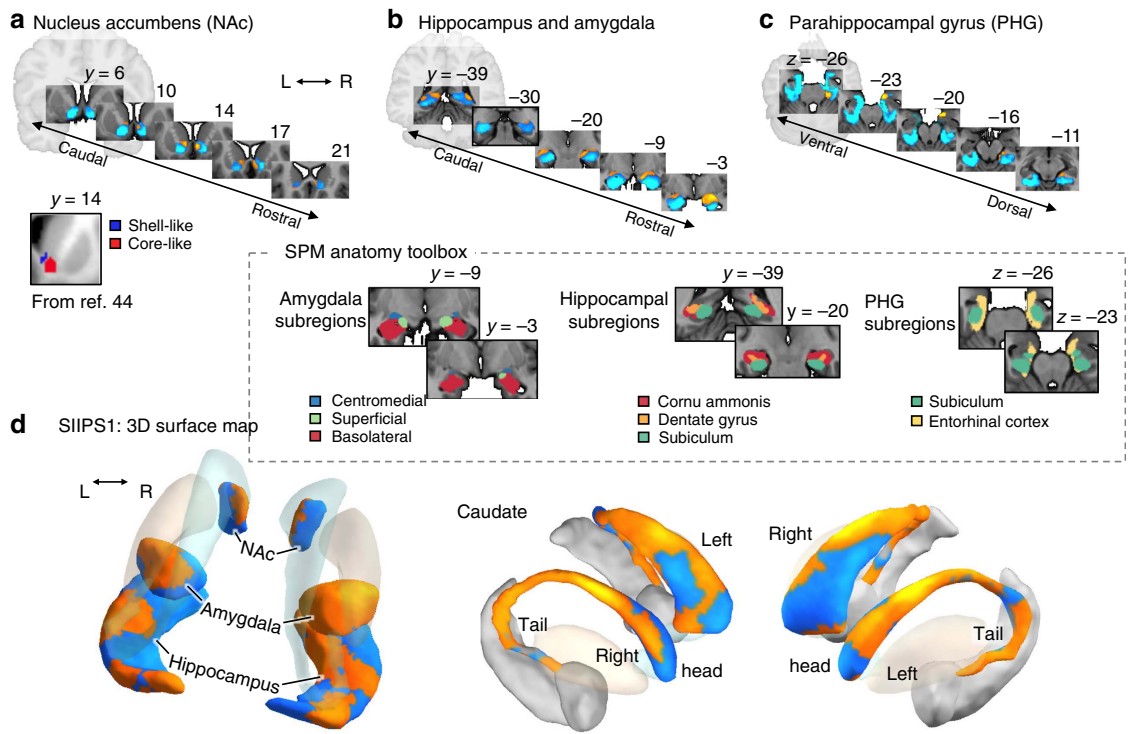

**Figure 3 | Unthresholded patterns of SIIPS1 predictive weights for some regions-of-interest (ROIs).** The ROIs include the nucleus accumbens (NAc), hippocampus, amygdala, PHG and caudate. The unthresholded pattern map used to make predictions included both positive and negative weights in each region, suggesting more complex, fine-grained mapping between these regions and pain. (**a**) Serial coronal views of the predictive weights within the NAc, showing positive predictive weights in a shell-like region and negative weights in a core-like region, as identified in a previous fMRI-based parcellation study[44]. Differential roles of the NAc shell (pro-pain) versus core (anti-pain) subdivisions have been shown in animal literature[15,46]. (**b**) Serial coronal views of the hippocampus and amygdala ROIs. Positive weights are apparent in the superficial and central subdivisions in the amygdala (as defined by ref. 47), and negative weights is the laterobasal group. A recent meta-analysis found that the superficial sub-region is often reported in experimental pain studies[51]. In the hippocampus, positive weights were found in some areas covering cornu ammonis and dentate gyrus[47], and also near caudate tail. (**c**) Serial axial views of the PHG ROI show positive weights in the entorhinal cortex (as defined by ref. 47) and a peri-amydaloid areas, and negative weights in other parahippocampal areas. (**d**) three-dimensional surface map of the un-thresholded SIIPS1 pattern for the ROIs. The pattern showed differential roles of caudate tail (positive) versus head (largely negative, but mixed), as suggested in animal[49] and metaanalysis studies[50], which associate caudate tail with stable, learned stimulus value and sensorimotor functions, and caudate head with more flexible, context-dependent stimulus value.

may correspond to the NAc core negatively predicted pain, corroborating similar earlier human[44,45] and animal studies[15,46]. In addition, an area covering the superficial and central subdivisions of the amygdala[47], in particular the right one[48], predicted increased pain, whereas activity in the basolateral subdivision[47] predicted reduced pain, paralleling animal literature[7,48]. Likewise, a part corresponding to the caudate tail associated with stable, learned values in animals[49] and sensorimotor associations in meta-analysis[50] show positive weights for pain, whereas a more anterior part of the caudate (that is, caudate head) associated with more context-dependent, flexible stimulus values[49,50] show mixed, but largely negative, weights. The SIIPS1 also showed fine-grained patterns of differential contributions in the parahippocampal gyrus (PHG) and hippocampus (Fig. 3b–d).

These patterns of predictive weights are consistent with recent findings in animal literature[7,15,46,48,49], suggesting that the topography we identified here could inform reverse translational approaches. In addition, the patterns within the amygdala, caudate and other regions build on recent meta-analyses that found, for example, superficial amygdala activation in experimental pain but implicated basolateral amygdala in chronic pain[51]. Interestingly, these topographical distinctions within brain regions are not at all apparent in univariate analyses (Supplementary Figs 4 and 5), suggesting that the multivariate

approach provides finer-grained and more sensible patterns related to pro- and anti-pain subregions. This high sensitivity of the multivariate pattern maps could be particularly useful for bridging the gap between the study of pain in humans and non-human animals.

**Joint predictive performance of the SIIPS1 and NPS.** To evaluate SIIPS1's predictive performance, we quantified the joint contributions of the SIIPS1 and the NPS in predicting trial-by-trial pain ratings. We used a multilevel general linear model to assess the unique and shared contributions of the SIIPS1 and the NPS to pain. We first conducted the analyses on the training data sets (Studies 1–4) using leave-one-participant-out cross-validation. This cross-validation procedure derives a pattern map from all training participants, except one out-of-sample participant, which is used to test the variance in pain explained by the brain pattern responses. The training and testing process is iterated until each participant is tested exactly once. We then conducted the same analyses on testing data sets (Studies 5–6) that were not included in the SIIPS1 pattern training. These analyses provide an unbiased test of how well the SIIPS1 captured fluctuations in pain above and beyond the NPS. In addition, to provide a preliminary examination of the SIIPS1's specificity

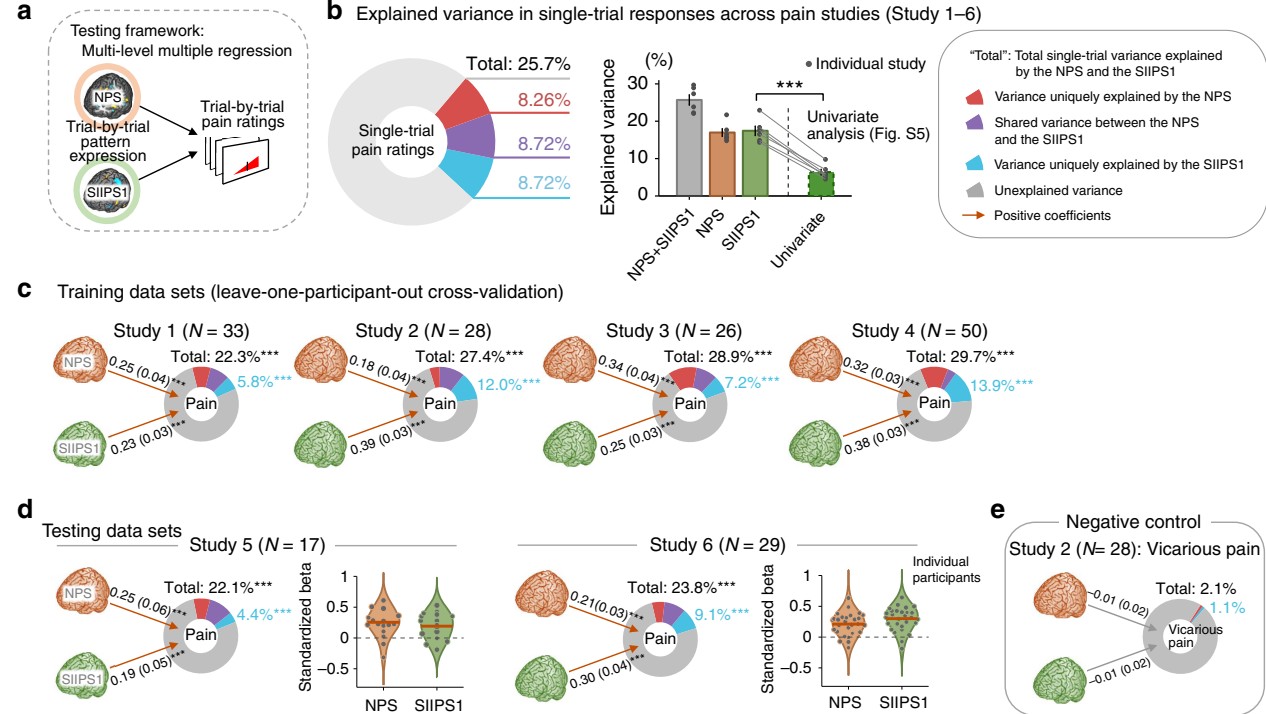

**Figure 4 | Joint contributions of two brain signatures to pain (predicting single-trial pain ratings).** (**a**) We estimated the unique and shared contributions to pain of the NPS and the SIIPS1 using multilevel general linear model. The trial-by-trial responses of the NPS and the SIIPS1 were independent variables, and trial-by-trial pain report was the outcome variable. (**b**) The pie chart and the bar plots show mean explained variance across six pain studies and dots on the bar plots show mean explained variance for each study. 'NPS + SIIPS1' indicates total variance explained by the NPS and the SIIPS1, and the 'NPS' and 'SIIPS1' indicates variance explained by the NPS and SIIPS1 separately. 'Univariate' indicates variance explained by the univariate map alone (for the univariate map; see Supplementary Fig. 5). Grey lines between 'SIIPS1' and 'Univariate' connect the same study, demonstrating that the univariate models consistently explain less variance in pain ratings than the multivariate models in cross-validated analyses. ***$P < 0.001$, one-sample $t$-test, which treats study as a random effect. (**c**) Regression coefficients (and s.e.m. in parenthesis) and explained variance by the NPS and the SIIPS1 in training data sets (Studies 1–4). Cross-validated signature response (leave-one-participant-out cross-validation) was used in the analysis. (**d**) Results from the testing data sets (Studies 5–6). Violin plots show the distribution of standardized $\beta$-coefficients for the NPS and SIIPS1 response using kernel density estimation and the red horizontal lines indicate Empirical Bayes weighted mean coefficients. (**e**) Results from the negative control data set (vicarious pain task with the same subjects of Study 2; see Methods). ***$P < 0.001$; Bootstrap (10,000 iterations) and permutation (5,000 iterations) tests were used for significance testing of regression coefficients and explained variance, respectively. For more details, see Methods.

to pain, we also tested the SIIPS1 and the NPS on a negative control data set, a vicarious pain task performed by participants in Study 2 (ref. 35).

As shown in Fig. 4, both in training (Studies 1–4) and testing data sets (Studies 5–6), the SIIPS1 and the NPS each made unique, significant contributions to predicting pain on individual trials, but no significant contributions to non-painful, aversive experience (vicarious pain induced by pictures[35]; for details, see Methods), demonstrating the SIIPS1's sensitivity and specificity for pain. The variance explained was comparable in magnitude for the SIIPS1 and the NPS. For the training data sets, the mean cross-validated regression coefficients (standardized) were $\hat{\beta}_{\text{SIIPS1}} = 0.312 \pm 0.040$ (mean ± s.e.m.) and $\hat{\beta}_{\text{NPS}} = 0.273 \pm 0.037$ (for Studies 1–4, $t$-values ranged from 7.73 to 20.0 for the SIIPS1 and from 5.22 to 9.98 for the NPS, all $P < 0.001$, multi-level general linear model, $N$ ranged from 26 to 50 depending on the study). The proportion of unique variance explained in single-trial pain was $9.71\% \pm 1.91\%$ for the SIIPS1 and $9.22\% \pm 1.90\%$ for the NPS. For the testing data sets, the mean standardized regression coefficients were $\hat{\beta}_{\text{SIIPS1}} = 0.246 \pm 0.054$ and $\hat{\beta}_{\text{NPS}} = 0.233 \pm 0.021$ (for Studies 5 and 6, $t$-values were 3.79 and 8.64 for the SIIPS1, and 4.70 and

6.46 for the NPS, all $P < 0.001$, multi-level general linear model, $N = 17$ and 29). The proportions of unique variance explained were $6.73\% \pm 2.33\%$ for SIIPS1 and $6.34\% \pm 0.86\%$ for the NPS. Permutation test results showed that, controlling for the NPS, the variance explained by the SIIPS1 was significant across each of the six studies individually (all $P < 0.001$). The total variance in single-trial pain ratings explained by the two fMRI signatures ranged from 22.1% to 29.7% across studies. This yielded ∼80% classification accuracy in discriminating high versus low pain for single trials (top 30% versus bottom 30% of trials) and over 94% accuracy when 4 or more trials are averaged together (Supplementary Fig. 6).

**Comparison with predictions based on univariate analysis.** For comparison, we also conducted a univariate analysis and used it as a decoding model. As in studies that use 'encoding–decoding' models[52], we estimated the regression coefficients for pain 'encoding' in each voxel separately based on voxel-wise general linear model and inverted the model to make predictions in out-of-sample individuals. To yield a single predicted pain value for each test image, we averaged the predictions from each individual voxel in the standard analysis and compared its predictive

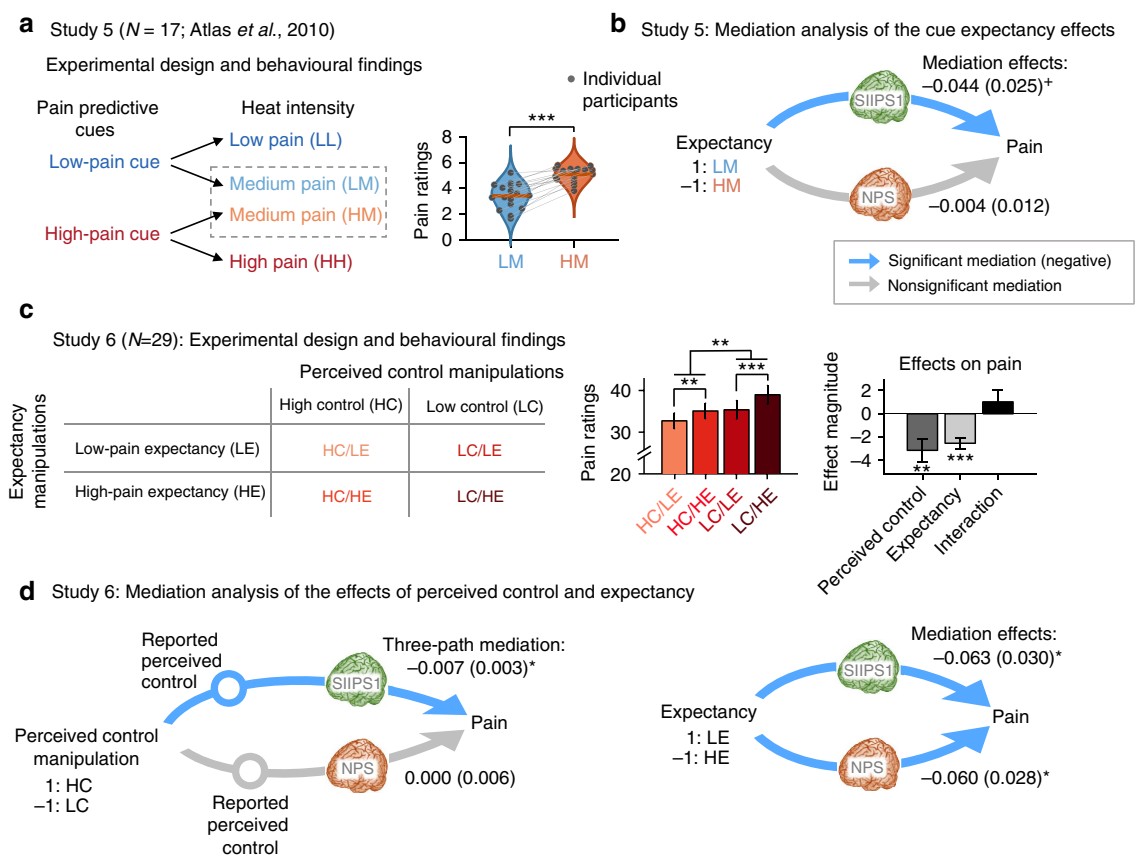

**Figure 5 | Mediation of psychological pain modulation (Studies 5–6).** (**a**) Study 5's experimental design and the behavioral findings: In Study 5, the expectancy level was manipulated by two different cues associated with two levels of heat intensity (high and low). After participants learned the cue-heat intensity association, the low-pain cue was followed by a low (LL trial type) or medium pain (LM) with 50% chance and the high-pain cue was followed by a medium (HM) or high pain (HH) with 50% chance. Violin plots show participants' averaged pain ratings for two medium pain conditions (LM and HM) and grey lines connect the same individuals' pain ratings. (**b**) The significant expectancy effects on pain were mediated by the SIIPS1, not by the NPS. The path coefficients and s.e.m. (in parenthesis) for the mediation effects (Path $a \times b$) are reported here. (**c**) Study 6's experimental design and the behavioral findings: in Study 6, we manipulated the levels of perceived control and expectancy with a 2-by-2 design (for more details, see Methods). The bar plots show participants' averaged pain ratings for four different experimental conditions and the right ones show the multilevel general linear model results (beta coefficients). Error bars represent s.e.m. (**d**) The perceived control effects were mediated only by the SIIPS1 (that is, not by the NPS) and the expectancy effects were mediated by both the SIIPS1 and the NPS. The path coefficients and s.e.m. (in parenthesis) for the mediation effects (Path $a \times b$) are reported here. For more detailed methods and statistics of path coefficients, see Methods and Supplementary Table 5. $^{+}P < 0.05$, one-tailed; $^{*}P < 0.05$, $^{**}P < 0.01$ and $^{***}P < 0.001$, two-tailed. Significance tests in this figure include paired $t$-test, multi-level mediation analyses (bootstrap test) and multi-level general linear model (bootstrap test).

accuracy with the multivariate results. The univariate map, shown in Supplementary Fig. 5, was similar to the multivariate map in most brain regions, but was smoother and did not show the fine-grained distinctions within regions described above. As shown in Fig. 4b, the univariate map explained significantly less variance than the multivariate SIIPS1 model in each of the six studies we tested. Across studies, the average variance in single-trial pain explained by the SIIPS1 was 17.4%, compared with 6.4% for the univariate map (mean difference $= 11.1\%$, $t(5) = 15.04$, $P < 0.0001$, paired $t$-test, $N = 6$ studies). The same pattern was found when we controlled for NPS responses. The SIIPS1 uniquely explained 8.7% of single-trial variance above and beyond the NPS, compared with 4.7% for the univariate map (mean difference $= 4.1\%$, $t(5) = 4.5$, $P < 0.01$, paired $t$-test, $N = 6$ studies).

**Mediation of psychological pain manipulations.** The NPS tracks pain accurately, but did not capture placebo effects[32], the effects of cognitive regulation[17] and effects of perceived control[34] on pain in previous studies. Given that the SIIPS1 predicts pain likely

to emerge from endogenous cerebral processes, it may be sensitive to pain modulation induced by psychological interventions. To examine this possibility, we conducted mediation analyses in two test data sets (Studies 5–6), providing an unbiased test of whether the SIIPS1, NPS or both mediate effects of expectancy and perceived control.

Study 5 (a re-analysis of a published data set[53]) examined expectancy effects. In a training phase, participants ($N = 17$) were told that one auditory cue was predictive of 'high pain' and the other a 'low pain' cue. These instructions were reinforced by conditioning to high- and low-intensity noxious heat, respectively, with intensities calibrated for each person. During a test phase on new skin sites, the high-pain cue was followed by high- or medium-intensity painful heat (50% probability of each) and the low-pain cue was followed by low or medium heat (50% each) (Fig. 5a). In an analysis of medium-intensity trials only, cues strongly biased pain reports towards the cued values (ratings for low-pain cue $= 3.42 \pm 0.24$ (mean $\pm$ s.e.m.) and for high-pain cue $= 5.08 \pm 0.14$, $\hat{\beta} = -1.73$, $t(16) = -10.2$, $P < 0.0001$, paired $t$-test, $N = 17$).

Using mediation analyses, we tested whether SIIPS1 and/or NPS responses mediated cue effects on pain (controlling for the other signature).

In Study 6 (a novel study), we crossed two psychological manipulations in a $2 \times 2$ factorial design. First, participants ($N = 29$) performed an operant 'symbolic conditioning' task found to affect pain and autonomic responses in previous work[54]. In the conditioning phase, options (denoted by abstract symbols) were probabilistically reinforced with visual feedback (thermometers) symbolizing high or low pain (Fig. 5c). The high-pain option was reinforced with high-pain feedback (50% of trials) or low-pain feedback (50%) and the low-pain cue was associated with high- and low-pain feedback on 20% and 80% of trials, respectively (Fig. 5c). All participants successfully learned these associations before scanning. During the in-scanner test, unbeknownst to participants, low or high temperatures were delivered with 50% probability for all the cue types (see Methods for more details). Second, participants did not have control over the option chosen on all trials. In high control (HC) blocks, participants chose which option was selected, whereas in low control (LC) blocks, choices were made by a computer. Thus, participants experienced noxious stimuli of equivalent intensity after choosing high- and low-expected pain options in both high perceived control and low perceived control blocks.

As shown in Fig. 5c, pain ratings for the experimental conditions were $32.7 \pm 1.9$ (mean $\pm$ s.e.m.) for the HC and low-pain expectancy (HC/LE) condition, $35.1 \pm 1.8$ for the HC and high-pain expectancy (HC/HE) condition, $35.4 \pm 2.2$ for the LC/LE condition and $39.0 \pm 2.2$ for the LC/HE condition. Both low (versus high) pain expectancy and high (versus low) perceived control resulted in strong, additive reductions in pain ($\hat{\beta} = -2.53$, $t(28) = -4.94$, $P < 0.0001$ for expectancy and $\hat{\beta} = -3.13$, $t(28) = -3.08$, $P = 0.005$ for perceived control, multi-level general linear model, $N = 29$). The expectancy × control interaction was not significant ($\hat{\beta} = 1.03$, $t(28) = 1.02$, $P = 0.32$, multi-level general linear model, $N = 29$).

In the multi-level mediation models, psychological manipulations (for example, low versus high-pain cues) were included as predictors (X), trial-by-trial pain ratings constituted the outcome variable (Y) and trial-by-trial SIIPS1 and NPS responses during pain were included as mediators (M). For the perceived control manipulation in Study 6, we tested a three-path mediation[17] involving serial associations between the control manipulation (X), self-reported perceived control (M1, the first mediator), trial-by-trial responses of the SIIPS1 and the NPS (M2, second-stage mediators), and pain ratings (Y). We coded anti-pain conditions (that is, LE and HC conditions) as 1 and pro-pain conditions as $-1$ (that is, HE and LC conditions) for X's, so that mediation effects were expected to be negatively signed.

As shown in Fig. 5b,d, the SIIPS1 partially mediated the effects of all three psychological manipulations on pain: for expectancy cues in Study 5, $\hat{\beta}_{\text{Path } a \times b} = -0.044$, $z = -1.79$, $P < 0.05$, one-tailed; for expectancy cues in Study 6, $\hat{\beta}_{\text{Path } a \times b} = -0.063$, $z = -2.11$, $P < 0.05$, two-tailed; and for perceived control in Study 6, three-path mediation $\hat{\beta} = -0.007$, $z = -2.08$, $P < 0.05$, two-tailed, multi-level mediation analyses with bootstrap tests, $N = 17$ for Study 5 and 29 for Study 6. Paths a and b—the cue effects on signature response and the relationship between signature response and reported pain, respectively—were individually significant in many cases, but did not always show significant effects even when the mediation effects (Path $a \times b$) were significant (Supplementary Table 5). This is a common phenomenon in multilevel mediation analyses when Paths a and b covary[53,55], indicating heterogeneity in the functional relationships involved. By contrast, the NPS showed more limited evidence for mediation of

psychological effects. It significantly mediated expectancy effects only in Study 6 ($\hat{\beta}_{\text{Path } a \times b} = -0.060$, $z = -2.16$, $P < 0.05$, two-tailed, multi-level mediation analyses with bootstrap tests, $N = 29$), but not expectancy in Study 5 or perceived control in Study 6. However, in Study 5, the NPS did respond more strongly to the high-pain versus low-pain cues ($\hat{\beta}_{\text{Path } a} = -0.273$, $z = -2.81$, $P < 0.01$, two-tailed, multi-level mediation analyses with bootstrap tests, $N = 17$).

These mediation results suggest that the SIIPS1 captures functionally meaningful variation in pain ratings as modulated by predictive cues (probably via expectations[54]) and perceived control. Thus, the SIIPS1 is likely to be influenced by psychological, 'top-down' influences on pain in ways that are not well captured by the NPS[17,32]. To see the full details of the mediation results, please refer to Fig. 5 and Supplementary Table 5.

## Discussion

In this study, we developed a multivariate fMRI signature, SIIPS1, predictive of variations in pain ratings after removing effects of stimulus intensity and nociceptive pain-related brain activity. The SIIPS1 was predictive of trial-by-trial pain ratings above and beyond variations in noxious stimulus intensity, suggesting that SIIPS1 reflects endogenous cerebral contributions to pain independent of nociceptive input to the brain. The signature included weight patterns that were consistent across individuals in a number of brain regions. It includes negative weights ('anti-pain' effects) in several regions related in previous studies to motivational value (vmPFC and NAc)[14,56], context and memory (for example, hippocampus and para-hippocampus)[57], and cognitive context (dlPFC)[58]. The SIIPS1 also included positive weights ('pro-pain' effects) in regions that receive nociceptive input (including the operculum, insula and cingulate cortex)[23] and frontal regions associated with higher-level cognitive processes (for example, dmPFC).

In addition, the relatively large sample combined with the multivariate analysis technique revealed fine-grained mapping of 'pro-pain' (positive weights) and 'anti-pain' (negative) sub-regions within pain modulatory regions, providing a more detailed characterization of pain-associated processes than has previously been available. For example, the unthresholded pattern of predictive weights within the SIIPS1 revealed that activity in a NAc shell-like region predicted increased pain and a core-like region predicted reduced pain, paralleling findings in human[44,45] and animal studies that have associated the shell with increased pain[46] and core with reduced pain[15]. In the amygdala, the activity in superficial and the central nuclei of the amygdala showed positive weights, whereas the basolateral subdivision of the amygdala showed negative weights, consistent with animal findings[7,48] and human neuroimaging literature[51]. The SIIPS1 mediated the effects of three psychological manipulations of pain from two independent studies, including two different expectancy manipulations and one perceived control manipulation.

An important contribution of the current study is in characterizing the relationship of nociception-independent regions in the prefrontal cortex and striatum with pain on one hand and with psychological interventions on the other. Recent studies suggest that they may play important roles in both acute and chronic pain. For example, although the NAc has not often been reported as being related to pain or regarded as a core pain system in the brain[23], emerging evidence from both human and animal studies suggests that the NAc plays critical roles in shaping affective and motivational value of pain[10,11,14], pain relief[59], pain-related behaviours[14,59] and chronic pain conditions[8,16,46]. Similarly, the SIIPS1 includes other brain

regions such as PHG and vmPFC, which are not often regarded as core pain processing regions and are not implicated in sensory aspects of pain, but have been implicated in different aspects of pain including chronic pain[5,6,9,15] and pain modulation[60–62].

Unlike previous studies, the current study provides a precise specification of the joint contributions of these nociception-independent regions, which can serve as an assay that can be easily shared and tested across different studies and laboratories. Although many of the brain regions we identified in this study have been previously reported, the previous studies are not consistent in their precise locations and in the direction of the effects. In addition, studies usually consider one brain region (or voxel) at a time or, in some cases, consider isolated pairs. This practice does not tell us what each brain region really 'does', because its functional roles may depend on activity in other regions. For example, some brain regions may play a secondary role that is indirect and mediated by other regions, and others may play a role that is masked by opposing effects of other, correlated regions and emerges only when controlling for it. In the current study, we provide a predictive map that specifies a precise set of locations and their relative contributions to pain controlling for other brain regions. This multivariate model reveals a fine-grained brain-pain mapping that is not redundant with previous univariate results (Fig. 3 and Supplementary Fig. 4), is substantially more predictive (Fig. 4b) and, unlike most previous neuroimaging findings, can be used prospectively to test interventions and patient populations in new studies (Figs 4 and 5).

Interestingly, dlPFC, vmPFC, NAc, amygdala, PHG and hippocampus were largely associated with less pain on average in the studies tested here, which may indicate a regulatory role based on contextualization of pain, as suggested by previous work[3,15,17,60,63]. However, these regions may play variable roles in pain modulation depending on individual differences and the cognitive context; for example, parahippocampal regions have been associated with anxiety-related pain increases in some studies[6,60]. Likewise, vmPFC activation is often associated with reduced pain in healthy controls[11,41], but it has also been associated with pain catastrophizing[63] and increased pain in chronic pain patients[43]. Our analyses also revealed some variation across studies (Supplementary Fig. 2) and individual differences in the relationship between increase or reduction in pain and the SIIPS1 (for example, nonsignificant Paths $a$ and/or $b$, but significant Path $a \times b$ in the mediation of the expectancy effects; Supplementary Table 5). In addition, a close examination of the SIIPS1 pattern revealed differences in the local pattern of effects across sub-regions (for example, across NAc shell-like versus core-like regions, superficial and central versus basolateral amygdala, and differences across portions of the PHG and hippocampus; Fig. 3), suggesting fine-grained structure of differential contributions (anti-pain or pro-pain). A full analysis of how these regions and their sub-regions may contribute differentially to pain in different experimental contexts, populations (for examples, patient groups) and individuals is beyond the scope of this study. The study does, however, lay a foundation for the future characterization of these effects.

In this regard, there are several concrete benefits to the signature pattern (or patterns if constituent local patterns are considered) we reported here. First, they identify multivariate patterns that have been optimized to explain pain more strongly than region-of-interest averages or univariate maps, and do so while controlling for the influences of other brain regions (due to the multivariate nature of the analysis). This is important because all the brain regions discussed contain neurons that participate in multiple, distinct functional circuits. Therefore,

identifying pain-predictive patterns provides measures of activity in representations more strongly linked to pain-relevant circuits than using region-of-interest averages. Second, because it was constructed to generalize across participants, the SIIPS1 (and its constituent local patterns) can be tested prospectively in future studies to further characterize its performance across experimental contexts and populations. Third, in conjunction with the NPS, SIIPS1 provides quantitative estimates of the activation intensity of at least two neurophysiological processes linked to pain, one peripheral stimulus-intensity dependent and the other endogenous and stimulus-intensity independent. Finally, both the NPS and SIIPS1 provide quantitative, physiological targets for pain interventions. Identifying neurophysiological targets for interventions is a major strategy for both validating the interventions and providing mechanistic insights into how they work[64]. For example, in neurological diseases such as Alzheimer's, researchers have developed named, diagnostic brain signatures that provide indicators of central pathology (for example, SPARE-AD for Alzheimer's[38]). These signatures do not replace clinicians' assessments, but inform clinical assessments by providing objective evidence for pathology and brain targets for interventions. Importantly, treatment development efforts have mainly focused on interventions for peripheral or central nociceptive processes—but the success or failure of such interventions may also depend on their effects on the non-nociceptive brain systems we identified here[62,65]. Therefore, the brain signature we developed here could help provide new brain targets for pain evaluation and treatment

There are some issues that could benefit from further discussion here. First, in addition to different psychological manipulations having different effects (which remains to be more fully characterized in future studies), even the same psychological treatments may have variable effects depending on details of the manipulation and population studied. For example, expectancy manipulations in Studies 5 and 6 were mediated by the SIIPS1, even though the SIIPS1 did not always increase with high-pain versus low-pain cues. Conversely, expectancy effects were partially mediated by the NPS in Study 6 but not Study 5, in spite of significant Paths $a$ and $b$ (cues to pattern responses and pattern response to pain, respectively). Such differences may be related to the strength and durability of conditioning or the depth of expectation, which may affect brain processes differentially even when they have similar effects on pain reports. This presents a challenge for future work, but also points to an opportunity to use these brain signatures to differentiate psychological treatments at a brain level even when they look similar at a behavioural level. In addition, for the same reason, we do not expect the SIIPS1 to be the one common mediator of all different types of top–down pain regulation. Therefore, the SIIPS1 should be considered as one candidate signature (thus, the '1' in the name); other brain patterns may mediate other types of 'top–down' pain regulation effects.

Second, as shown in Fig. 4, the two fMRI signatures that we tested (that is, the SIIPS1 and the NPS) together explained around a quarter (25.7%) of the total variance in single trial-level pain ratings, which can achieve 80.3% classification accuracy in discriminating high pain from low pain trials (top 30% versus bottom 30% of trials; Supplementary Fig. 6). At the single-trial level, this is a modest proportion of variance explained, but the single trial-level data are very noisy and contain many sources of unexplained variance, such as inter-individual, inter-study variations and measurement errors. As Supplementary Fig. 6 shows, if we average over several trials, the relationships between average signature responses and average pain increase dramatically; for example, averaging over 25 trials, the signatures explain 89.4% of the variance in reported pain experience, with

near-perfect classification accuracy for high versus low pain. In Supplementary Fig. 6, we grouped trials with similar pain ratings, which still provides an unbiased measure of accuracy, because the model has no prior information about which groups of trials are more painful than others. The same principle applies to any *a priori* grouping of trials; for example, groups of trials tested under different treatments or groups of trials administered to normal versus hypersensitive skin.

Third, even though we trained the signature to be predictive of variation in pain after removing the effects of nociceptive input and related brain processes, the SIIPS1 still contains brain regions that respond to nociceptive input (for example, parts of insula and cingulate cortex). As explained above, this is sensible if endogenous variations in these nociceptive regions contribute to pain beyond encoding noxious input intensity. However, it might still be interesting to develop a brain signature that is purely non-nociceptive—that is, shows no response to nociceptive input at all—but is still predictive of pain experience. Figure 2c provides promising candidate brain regions, including NAc, dm/vmPFC, hippocampus, temporal pole and precuneus, which are unresponsive to nociceptive input and predictive of residual pain ratings. Further examination of their roles in pain and the development of pain signatures based on those brain regions could be an interesting future direction.

Fourth, although we aimed to identify a signature that can be applied to new data from new individuals (that is, population-level model) in this study, idiographic models could also be useful for pain prediction[66]. Idiographic modelling approaches might be able to model the effects of individual differences on pain (for example, personal history and memories related to pain), revealing individualized neural bases for pain perception. In addition, we modelled the SIIPS1 using only spatial pattern information, yielding a static template. In the future, dynamic modelling approaches can also be used to take into account the temporal dynamics and information flow among brain regions.

Overall, our study and SIIPS1, together with the NPS, provide new ways of understanding and evaluating the neurobiological components of pain. These a priori brain signatures can be prospectively used to assess pain contributions that are nociceptive and beyond nociceptive in new individuals, and therefore provide a step towards a quantitative assessment of the multiple components of pain.

## Methods

**Participants.** The study included a total of 183 healthy participants from 6 independent studies, with sample sizes ranging from $N = 17$ to $N = 50$ per study. Descriptive statistics on the age, sex and other features of each study sample are provided in Supplementary Table 1. Participants were recruited from New York City and Boulder/Denver Metro Areas. The institutional review board of Columbia University and the University of Colorado Boulder approved all the studies, and all participants provided written informed consent. Preliminary eligibility of participants was determined through an online questionnaire, a pain safety screening form and an MRI safety screening form. Participants with psychiatric, physiological or pain disorders, neurological conditions and MRI contraindications were excluded before enrolment.

**Procedures.** In all studies, participants received a series of contact-heat stimuli and rated their experienced pain following each stimulus. Data from Studies 1–6 have been used in previous publications (see Supplementary Table 1 and ref. 66); however, the analyses and findings reported here are novel and have not been published elsewhere, and the analyses on psychological pain modulation effects in Study 6 have not appeared in any prior publications. The number of trials, stimulation sites, rating scales and stimulus intensities and durations varied across studies, but were comparable; these variables are summarized in Supplementary Table 2. Each study also comprises a specific psychological manipulation, such as cue-induced expectation and placebo treatment. In the studies included in the training data sets (Studies 1–4), we focused only on residual pain ratings (ratings after removing noxious stimulus intensity and the NPS response) irrespective of the study-specific psychological manipulations. In the studies in the testing data sets (Studies 5–6), we carried out mediation analyses using the

study-specific psychological manipulations, including expectancy induced by cues paired with verbal instructions and conditioning (Study 5) or cues associated with different probabilities of receiving low pain (Study 6), and perceived control (Study 6).

**Cognitive self-regulation in Study 1.** On some runs (third and seventh runs among nine runs) of Study 1, participants implemented a cognitive self-regulation strategy directed at either increasing ('Regulate-up') or decreasing ('Regulate-down') pain. The strategy was similar to reappraisal procedures commonly used to 'rethink' responses to images and events, which also involve a mix of mental imagery and subvocalized narrative. This intervention was designed to target both sensory and affective components of pain based on effective self-regulation strategies used in prior pain studies. For the full instructions, see ref. 17.

**Cue-induced expectancy in Study 2.** Study 2 included three levels of predictive visual cues that corresponded to three levels of stimulation and each thermal stimulation was preceded by one of the three cues. The cues and stimulation were crossed with each other; these predictive cues were orthogonal to the intensity of stimulation. Before the main experiment, participants completed a short training session with an explicit learning task where they learned the levels of the three cues that were later presented in the scanner. They also underwent two runs of a conditioning task in the scanner where the participants learned the association between the cues and the level of stimulation. During the main experiment (total nine runs and each run had nine trials), participants received fully crossed pairs of cues and stimulus intensity.

**Masked emotional faces in Study 3.** At the start of each trial, a square appeared in the center of the screen for 50 ms, followed by a pair of faces from the Ekman set[67]. An emotional expression (Happy or Fearful) was presented for 33 ms, masked by a neutral face presented for 1,467 ms. Face cues were evenly crossed with temperature. For more details, see ref. 41.

**Cue- and placebo-induced expectancy in Study 4.** Study 4 used two levels of predictive visual cues that were associated with two levels (high and low) of heat intensity. One cue was always followed by a low pain (46 °C) or a medium pain (47 °C) stimulus (with 50% chance) and the other cue was always followed by a medium pain (47 °C) or a high pain (48 °C) stimulus (with 50% chance). Participants were not informed about these associations before the experiment and therefore needed to learn the associations during the experiment. The participants chose the cue that they thought was associated with less pain. Combined with this cue manipulation, Study 4 administered the heat stimuli on two different types of skin sites: skin sites that had been treated with a placebo analgesic cream (placebo condition) or had not been pretreated (control condition). With these two types of skin sites, the cue-learning task alternated between placebo and control runs in counterbalanced order across participants. For more details, see ref. 61.

**Cue-induced expectancy in Study 5.** Study 5 included two auditory cues that were associated with two levels (high and low) of heat intensity. First, a calibration procedure established 'low pain' (Level 2 on a 10-point visual analogue scale (VAS) anchored at 'no pain' and 'extreme pain') and 'high pain' (Level 8) intensities for each participant. Then, participants were instructed that one tone would be followed by low pain and the other would be followed by high pain. During fMRI scanning, each thermal stimulation was preceded by one of the two auditory cues. In the first two runs, the low-pain cue was always followed by a low-pain stimulation and the high-pain cue was always followed by a high-pain stimulation. These two conditioning runs served participants to reinforce verbal instructions. Next, there were six runs where low pain cue was followed by a low (LL) or medium pain (LM) with 50% chance and the high pain cue was followed by a medium (HM) or high pain (HH) with 50% chance (Fig. 5a). To maximize the expectancy effects for the medium pain stimuli, participants were not told that medium pain stimuli would be applied. In the mediation analyses of psychological modulation effects (Fig. 5b), we included only medium pain trials (LM and HM) to see the expectancy effects on the same intensity stimuli. For more details, see ref. 53. Cue assignment for high versus low pain was counterbalanced across participants.

**Perceived control and expectancy in Study 6.** Study 6 aimed to disentangle the relative contributions of perceived control and expectancy induced by certainty manipulations to pain experience in a 2-by-2 design (Fig. 5c). First, a calibration procedure established 'low pain' (Level 4 on a 10-point VAS anchored at 'no pain' and 'worst pain imaginable') and 'high pain' (Level 6) intensities for each participant. Then, in the main experiment, two levels of perceived control were induced by (1) allowing participants to choose between two visual cues (HC) versus (2) having them observe a cue choice made by a computer (LC). Second, two levels of expectancy were manipulated with two different types of cue pairs (80/20 cue pair versus 50/50 cue pair) associated with different probabilities of receiving high or low pain. In the 80/20 cue pair (LE), one cue was associated predominantly with

low pain (80% low-pain and 20% high-pain) and the other cue was associated with high pain (20% low-pain and 80% high pain). In the 50/50 cue pair (HE), both cues were associated with a 50% chance of receiving high and low pain. Cues were counterbalanced across participants.

The paradigm consisted of two phases. During the first phase, participants completed the instrumental learning task where they learned the associations between cues and particular painful heat outcomes using visual feedback in the form of a thermometer. We have referred to this as 'symbolic conditioning', because the reinforcers are symbolic indicators of pain, but there is no primary reinforcement (for example, no actually painful reinforcers)[54]. This learning phase consists of 2 runs and each run had 60 trials (1 run for the HC condition and the other for the LC condition). After the learning phase, we conducted the forced cue-choice task (six trials for each experimental condition across two runs) to see if the participants successfully learned the cue–outcome association. All participants correctly chose the cue associated with low pain in 84% of trials and the cue associated with high pain in 46% of trials, demonstrating probability matching to the frequencies of outcomes and indicating that participants successfully learned the cue–pain associations.

During the second phase, participants underwent the pain task with the actual painful heat stimuli in the MRI scanner. In this pain task, participants chose a cue or observed the cue choice and then received painful heat as feedback. This test phase consists of eight runs and each run included eight trials. Therefore, there were total 64 trials and 16 trials for each experimental condition of the 2-by-2 design. Unbeknownst to participants, low or high temperatures were delivered with 50% probability in all conditions, to avoid confounds between experimental manipulations and pain history[68]. Participants provided ratings for perceived control, expected (before pain stimulation) and actual pain ratings (after pain stimulation) on a 100-point VAS.

**Thermal stimulation.** In each study we delivered thermal stimulation to multiple skin sites using a TSA-II Neurosensory Analyzer or Pathways system (Medoc Ltd, Chapel Hill, NC) with a 16 mm Peltier thermode endplate. On every trial, after the offset of stimulation, participants rated the magnitude of the warmth or pain they had felt during the trial on a VAS or labelled magnitude scale. Other thermal stimulation parameters varied across studies, with stimulation temperatures ranging from 40.8 to 49.3 °C and stimulation durations from 10 to 12.5 s. All studies applied thermal stimulation to the glabrous skin of the left forearm and Study 2 additionally applied the stimulation to the dorsum of the left foot. See Supplementary Table 2 for location of stimulation sites, stimulation intensity levels, stimulation duration and the number of trials per subject.

**Vicarious pain task.** Study 2 (ref. 35) included two different 'pain' tasks, tested within-participants in separate sessions on different days to reduce any carry-over effects. The somatic pain task described above involved experiencing three levels (low, medium and high) of noxious heat and the vicarious pain task involved viewing images that contained painful events in others. Participants were asked to imagine that the injury occurring in the picture was happening to them and rate how much pain they might feel in that situation. We grouped pictures into three intensity levels based on prior norms[69] that were approximately matched on the intensity of negative affect ratings. The structure and timing of the vicarious pain task matched that of the somatic pain task. In a training session, three predictive cues were associated with three levels of vicarious pain stimuli. In the fMRI session, these three cues were fully crossed with the three levels of vicarious pain pictures and we analysed the relationships between brain activity and actual reported 'pain' experience (using the normative intensity levels as an instrument to induce appropriate variance and ensure balance in the stimulus intensities presented across time). Here we report relationships between the pain signatures (NPS, SIIPS1 and both combined) and reported vicarious pain intensity (Fig. 4e). A previous publication on the vicarious pain task showed no responses to the NPS[35]; thus, we expected this to serve as a negative control here, testing whether the SIIPS1 differentiated somatic from vicarious pain.

**Preprocessing of fMRI data.** Structural T1-weighted images were co-registered to the mean functional image for each subject using the iterative mutual information-based algorithm implemented in SPM and were then normalized to MNI space using SPM. SPM versions varied across studies (Studies 3 and 5 used SPM5; all other studies used SPM8; http://www.fil.ion.ucl.ac.uk/spm/). Following SPM normalization, Studies 3 and 5 included an additional step of normalization to the group mean using a genetic algorithm-based normalization[41,53,70]. In each functional data set, we removed initial volumes to allow for image intensity stabilization (see Supplementary Table 3 for number of initial volumes removed in each study). We also identified image-intensity outliers (that is, 'spikes') by computing the mean and s.d. (across voxels) of intensity values for each image for all slices to remove intermittent gradient and severe motion-related artefacts present to some degree in all fMRI data. To identify outliers, we first computed both the mean and the s.d. of intensity values across each slice, for each image. Mahalanobis distances for the matrix of (concatenated) slice-wise mean and s.d. values by functional volumes (over time) were computed. Any values with a significant $\chi^2$-value (corrected for multiple comparisons based on the more

stringent of either false discovery rate or Bonferroni methods) were considered outliers. In practice, <1% of images were deemed outliers. Each time point identified as outliers was later included as nuisance covariates in the first-level models.

Next, functional images were corrected for differences in the acquisition timing of each slice and were motion corrected (realigned) using SPM. The functional images were warped to SPM's normative atlas (warping parameters estimated from co-registered, high-resolution structural images), interpolated to $2 \times 2 \times 2 \, mm^3$ voxels and smoothed with an 8 mm full width at half maximum Gaussian kernel. This smoothing level has been shown to improve inter-subject functional alignment, while retaining sensitivity to mesoscopic activity patterns that are consistent across individuals[71].

**Single trial analysis except for Study 2 and 5.** For each study, except for Study 2 and 5, we employed the single trial or 'single-epoch' design and analysis approach[72]. We estimated single-trial response magnitudes for each brain voxel using a general linear model design matrix with separate regressors for each trial, as in the 'beta series' approach[73]. First, boxcar regressors, convolved with the canonical haemodynamic response function (HRF), were constructed to model cue (if any), pain stimulations (somatic or vicarious) and rating periods in each study. Then, we included a regressor for each trial, as well as nuisance covariates (for example, linear drift across time within each run; the six estimated head motion parameters ($x$, $y$, $z$, roll, pitch and yaw); indicator vectors for outlier time points identified based on their multivariate distance from the other images in the sample).

One important consideration in the single trial analysis is that trial estimates could be strongly affected by acquisition artifacts that occur during that trial (for example, sudden motion, scanner pulse artifacts and so on). For this reason, trial-by-trial variance inflation factors (VIFs, a measure of design-induced uncertainty due, in this case, to colinearity with nuisance regressors) were calculated and any trials with VIFs that exceeded 2.5 were excluded from the following analyses. For Study 3, we also excluded global outliers (trials that exceeded three standard deviations above the mean) and employed a denoising step based on principal component analysis during preprocessing to minimize artefacts.

**Single trial analysis for Study 2 and 5.** For Study 2 and 5, single trial analyses were based on fitting a set of three basis functions rather than the standard HRF used in the other studies. This flexible strategy allowed the shape of the modeled HRF to vary across trials and voxels. This procedure differed from that used in other studies included in the current study, mainly because it maintains consistency with the procedures used in the original publication[53]. For both Study 2 and Study 5, the pain period basis set consisted of three curves shifted in time and was customized for thermal pain responses based on previous studies[53]. To estimate cue-evoked responses for Study 5, the pain anticipation period was modelled using a boxcar epoch convolved with a canonical HRF. This epoch was truncated at 8 s, to ensure that fitted anticipatory responses were not affected by noxious stimulus-evoked activity. As with the other studies, we included nuisance covariates and excluded trials with VIFs > 2.5. In Study 5, we also excluded trials that were global outliers (those that exceeded 3 s.d. above the mean). We reconstructed the fitted basis functions from the flexible single trial approach to compute the area under the curve for each trial and in each voxel. We used these trial-by-trial area under the curve values as estimates of trial-level pain-period activity.

**Developing SIIPS1.** We developed the SIIPS1 using single-trial estimates of brain responses during individual epochs of noxious heat from 137 participants in Studies 1–4 (8,224 trials total; we removed non-painful trials in the signature development step, resulting in 6,740 images total and 50 trial images per person on average). We used two-step approach that consisted of individual- and group-level analyses.

For each individual, we first regressed out stimulus intensity and NPS response from single-trial estimates of brain activity and pain ratings. The NPS response was calculated using the dot product of a vectorized single-trial activation map with the NPS weights. We removed the effects of stimulus intensity (temperature) using a non-parametric method by creating indicator regressors for different levels of stimulus intensity (that is, temperature). This model effectively matches on stimulus intensity, as it removes the mean pain ratings and mean brain activity within each voxel for each stimulus intensity level. We also included the NPS response as an additional regressor to account for remaining variations in peripheral nociceptive input within-temperature to the degree possible. Thus, the residuals that we used to predict pain are orthogonalized with respect to the subspace that spans stimulus intensity effects (linear and nonlinear) and the NPS. Then, we used PCR[39] to predict residualized pain rating from the residualized single-trial whole brain activity to obtain stable predictive models with high-dimensional, collinear predictors.

After we obtained predictive maps for all individuals using PCR, we constructed a group map using precision-weighted average. For precision estimates, we calculated prediction–outcome correlation with tenfold cross-validation for each subject. Before calculating weighted averages, we normalized each participant's

PCR weights using s.d. of the weights to minimize differences in scales across studies. To examine which brain regions made reliable contributions to prediction across participants, we conducted weighted $t$-test (Fig. 1c). To capture common neural components across different types of top–down and endogenous cerebral influences on pain, studies in the training data included a heterogeneous set of psychological modulation tasks (or no psychological modulation) in the training data sets (Supplementary Table 2). For the graphical overview of the SIIPS1 development, see Supplementary Fig. 1.

**Testing the SIIPS1 on new data sets.** To test the SIIPS1's performance in independent testing data sets, we calculated the strength of pattern expression of the SIIPS1 (that is, signature response) using the dot product of a vectorized single-trial activation map with the SIIPS1 pattern weights, yielding a scalar value. In the multi-level general linear model and mediation analyses, we used the SIIPS1 response calculated from the single-trial beta images. For training data sets, we used a leave-one-participant-out test, which iteratively derives pattern maps from training data, except for one out-of-sample participant, and calculated the signature response for the out-of-sample participant.

**Correlation analyses with stimulus intensity.** We first obtained contiguous regions from the SIIPS1 that survived false discovery rate correction ($q < 0.05$) and were larger than 15 voxels (except for the right NAc, which has 7 voxels), resulting in 44 contiguous sub-regions. With these regions, we calculated correlations between brain regions' local pattern expression (using absolute pattern weights) and trial-by-trial noxious stimulus intensity for each participant. The reason we used absolute weight values in this analysis is to make correlation values easy to interpret; positive correlations mean positive relationships between the region's activation and stimulus intensity, and negative correlations indicate negative relationships between the region's activation and stimulus intensity. We then obtained mean correlations across six independent studies combining training and testing data sets ($N = 183$). For significance testing, the correlations were converted to $z$-values using Fisher's $z$-transformation and $t$-test was conducted on the Fisher's $z$-values across participants. Then, we corrected $P$-values for multiple comparisons using the Bonferroni procedure ($\alpha = 0.05/44 = 0.0011$).

**Univariate analyses.** To test the relative performance of the multivariate model to a univariate voxel-wise model in explaining variations in trial-by-trial pain ratings, we constructed a univariate map using an encoding–decoding approach. As in the multivariate analyses, the univariate analysis consisted of individual- and group-level analysis steps. For the individual-level analysis, we estimated $\beta$-coefficients (regression slopes) for each voxel based on the regression models that predicted each voxel's residualized brain activity from residualized pain ratings in training data sets, (Studies 1–4). We then constructed a group map by averaging the univariate maps for all individuals and thresholded the map (only for display) by performing a one-sample $t$-test with false discovery rate $q < 0.05$ (equivalent to voxel-wise $P < 0.0085$; Supplementary Fig. 5a). To decode pain in out-of-sample test participants, we inverted the model to make predictions for each test individual by treating the averaged $\beta$-coefficients as predictive weights and averaging the univariate predictions from each voxel across the brain into a single predicted pain value for each test trial. For the training data, we used a leave-one-participant-out cross-validation, which iteratively derives maps from training data excluding one out-of-sample participant and calculated predicted pain for each trial in the out-of-sample participant. We then summarized the predictive accuracy across all test individuals (that is, across folds in Studies 1–4 and across new individuals in Studies 5–6).

**Multilevel general linear model.** First, to quantify joint contributions of the SIIPS1 and NPS to pain (Fig. 4), we used multilevel general linear model, implemented with custom code written in Matlab (glmfit_multilevel.m; available at https://github.com/canlab/CanlabCore). The outcome variable was trial-by-trial pain ratings and the dependent variables included trial-by-trial pattern expression of the SIIPS1 and the NPS. To compare the two $\beta$-coefficients, the standardized values ($z$-scored pattern expression values across trials) were used for the dependent variables. We calculated the SIIPS1 response for the training data sets using a leave-one-participant-out cross-validation procedure. For significance testing, we used bootstrap tests, where two-tailed $P$-values were calculated based on the distributions of group-level regression coefficients estimated by randomly sampling (with replacement) the observations 10,000 times. In addition, the Empirical Bayes weighting procedure based on first-level variance estimates of $\beta$-coefficients was used in second-level analyses including bootstrap tests[74]. The variance explained ($R^2$) by the full and reduced models was calculated for each participant and then averaged for group-level estimates. For significance testing of $R^2$, we used permutation tests, where trial labels for the variable tested were randomly shuffled for each participant, and the group-level $R^2$ with the permuted data was estimated for each iteration. Then, two-tailed $P$-values were calculated based on the distributions of the group-level estimate of $R^2$ for the models of interest.

Second, the multilevel general linear model was also used to examine the effects of psychological interventions on pain ratings in Studies 5 and 6 (Fig. 5). In these analyses, the outcome variable was trial-by-trial pain ratings and the dependent variables were experimental conditions. For more details about the coding scheme of the dependent variables, please refer to the next section (Multilevel mediation analysis) or Fig. 5b,d.

**Multilevel mediation analysis.** To examine which combination of the SIIPS1 and the NPS mediates the effects of psychological intervention, multilevel mediation analyses were performed using the Mediation Toolbox written in Matlab (mediation.m for two-path mediation and mediation_threepath.m for three-path mediation[17]; available at https://github.com/canlab/MediationToolbox). The mediation analysis tests whether a covariance between two variables (X and Y) can be explained by one (M) or two intermediate variables (M1 and M2). More technical details on the two-path and three-path mediation analyses implemented in the Mediation Toolbox can be found in refs 17,75.

In the current study, a psychological manipulation for each study was entered as a predictor (X), trial-by-trial pain ratings were entered as an outcome variable (Y) and the trial-by-trial SIIPS1 and NPS responses were entered as mediators (M). For Study 5, low-pain cue followed by medium pain (LM) was coded as 1 and high-pain cue followed by medium pain (HM) was coded as $-1$. For Study 6's expectancy manipulation, cues associated with a 20% probability of receiving high-pain (80/20 cue pair) were coded as 1 (LE) and cues associated with a 50% probability of receiving high pain (50/50 cue pair) were coded as $-1$ (HE). For the perceived control manipulation in Study 6, high perceived control trials (making cue choices) were coded as 1 and low perceived control trials (observing cue choices) were coded as $-1$. For Study 6, stimulus intensity (temperature) was entered as a covariate. In addition, in the analysis for the perceived control, we included the self-reported perceived control as the first mediator (M1), and the SIIPS1 and the NPS were tested for the second mediators (M2) using three-path mediation. Bootstrap tests (10,000 iterations) were used for significance testing of mediation effects.

**Data availability.** The data that support the findings of this study are available from the corresponding author upon reasonable request.

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

## Acknowledgements

This work was funded by NIH R01DA035484 and R01MH076136 (T.D.W.), the Intramural Research program of the NIH's National Center for Complementary and Integrative Health (L.Y.A) and the VENI grant of the Netherlands Organization for Scientific Research (M.J.).

## Author contributions

C.-W.W. and T.D.W. analysed the data, interpreted the results and wrote the paper. C.-W.W., M.R. and T.D.W. contributed to Study 1 data. A.K., C.-W.W. and T.D.W. contributed to Study 2 data. L.Y.A. and T.D.W. contributed to Study 3 and Study 5 data. M.J. and T.D.W. contributed to Study 4 data. L.S. and T.D.W. contributed to Study 6 data. M.A.L. helped with data aggregation and statistical analyses. All authors contributed to the preparation and revision of the manuscript.

**Additional information**

**Competing financial interests:** T.D.W. and M.A.L. hold patent US 2016/0054409 'fMRI-based Neurologic Signature of Physical Pain (PCT/US14/33538)'. The remaining authors declare no competing financial interests.

