## [Peer Review File · Nature Communications]

PEER REVIEW FILE

Reviewers' Comments:

Reviewer #1 (Remarks to the Author):

Thanks to the authors for providing a thorough rebuttal to the issues raised in my first review. I think the work, including the additional analyses performed in the latest version, represent a comprehensive and methodologically robust investigation of the effects of psychologically mediated variables on brain activity and behaviour. From my perspective, I think the manuscript is suitable for publication, and represents a sufficiently novel advance for ncomms.

Reviewer #3 (Remarks to the Author):

The main thesis of this line of work is the notion that there exist static brain activity patterns across subjects that underlie perception. Here specifically the authors try to identify patterns for pain when the stimulus and related brain pattern are regressed out. The remaining signal and related pattern they call a new signature of pain independent of nociception. Thus supposedly unraveling brain elements the combination of which underlies pain perception independent of the stimulus, a very large claim indeed. Note no such similar claim exists for any other sensory modality and I strongly suspect that the present claim is not valid.

Obviously the relationship between brain location and function is a well-established fact. However, reducing this into a static template across all humans strongly harks of phrenology, and contradicts a large body of science repeatedly showing that perception, pain or otherwise, is dependent on past experiences, memories, and on individual history of life, and also contradicts the huge body of science showing that cortical synaptic properties and activity are constantly changing with experience.

The paper is very impressive regarding the massive computational technology that the authors employ. However the technology seems to have convinced the authors that they have discovered a new fundamental brain structural organization rule, where instead I think they are being misled by statistical tricks. There is also a loose use of semantics that creates the notion that something profound has been uncovered.

The final outcome of the analyses is an optimized weighted sum of all brain voxels (around 60,000 elements) to map the discrimination of pain levels across multiple studies. The concept is very simple. For example if we reduce the brain to 3 elements: visual cortex (V), pain cortex (P), and cognitive cortex (C). Then the solution of NPS could be something like: -1, 1, and 0 (weights for V, P, C), while after removing the painful stimulus it may become: -1, 0.5, 0.5. The latter becomes a crude version of their SIIPS1. The only difference between my toy NPS and SIIPS1 models and the one in the paper are the number of elements (60, 000 weighted elements). The latter provides enough degrees of freedom that simple tasks like discriminating between high vs. low pain can be obviously solved, even for new data sets. The solution is not due to the discovery of a useful pattern but relies on the vast availability of degrees of freedom.

The original formulation of NPS was based on whole-brain reverse mapping for noxious stimuli. In that paper the authors argued that the signature was not for nociception but reflected pain as NPS better correlated with reported pain than the stimulus. However, in subsequent studies they have in fact quickly discovered that NPS fails in very simple manipulations, like cognitive modulation. Thus NPS cannot be considered a "pain signature". Therefore, they have now derived a new signature and the old one is now dubbed "nociceptive" pain signature. A more modest and more straightforward explanation would be that NPS itself is a mixture of pain, nociception, and most importantly somatic stimulus magnitude as well as other non-specific perception/stimulus correlated brain activity. This argument is directly relevant to the introduction of the paper where now they argue (with no direct evidence) that NPS or "pain matrix" are nociceptive brain activity. This is not true as in all such studies pain, nociception, attention, heart rate changes etc. are all entangled in the outcome. Still I would accept that NPS reflects some optimized weighted GLM result regarding nociception/pain, where the optimization is specifically enables performing reverse inference. Within this view what is SIIPS1? Here the authors regress out stimulus intensity and NPS and search for brain signals that capture pain after removing these factors. At least in their original NPS paper, pain and stimulus were more than 80-90% correlated with each other. In such a case, regressing the stimulus leaves residuals that have minimal power. In fact what is guaranteed is that such residuals are dominantly noise. The authors of this paper contend that since their maps show statistically significant outcomes that capture data from out of training sets, these cannot be noise. This is false. There is plenty of opportunity for noise to attain predictable patterns that can be captured across studies. The simple example is that subjects move their head more for more painful stimuli and although very sophisticated motion correction paradigms are now available, it is very clear that motion cannot be completely removed from fMRI and also that very aggressive removal of motion would introduce artificial new patterns in the data. Similar uncontrolled artifacts can also creep in in such tasks for example by changes in heart rate, respiration, extent of anxiety, attention, urge to move or inhibition of urge to move, or actual movement etc. all of which are factors that change with pain intensity. So removing the stimulus intensity per se leaves all of these effects in the brain and none of these should be called pain specific signatures.

It remains unclear how much variance is available in the multiple tasks studied here, once the intensity of the stimulus is removed. In study 1 it seems that cognition is providing additional variance but I cannot discern this in the other studies. Importantly, although the authors show significant results, the

effect sizes are tiny and one could just as well interpret that obtained results have little biological significance (suggesting to me that they are capturing non-specific patterns).

It is also very clear from the presented results that the GLM and the calculated signatures are in fact essentially the same, after all the signatures are slightly weighted and averaged beta maps. I suspect a voxelwise correlation between NPS, SIIPS1 and GLM will show massive covariation.

The authors need to be far more open and frank regarding what they are doing and take a more objective view of these results. I have made similar comments to earlier versions of this paper but the authors have ignored them and simply submitted the paper to another journal. I am certain that just the statistical manipulations will impress enough reviewers to make this paper be published somewhere. However I urge the authors to seriously examine what it is they are creating and how much of the results are biologically meaningful.

Ms. No.: NCOMMS-16-14853-T

Response to Reviewers

We would like to thank the *Nature Communications* editors for the opportunity to respond to this review. Please find below our detailed, point-by-point response to the Reviewers' comments.

Reviewer #3

3-1 "The main thesis of this line of work is the notion that there exist static brain activity patterns across subjects that underlie perception. Here specifically the authors try to identify patterns for pain when the stimulus and related brain pattern are regressed out. The remaining signal and related pattern they call a new signature of pain independent of nociception. Thus supposedly unraveling brain elements the combination of which underlies pain perception independent of the stimulus, a very large claim indeed. Note no such similar claim exists for any other sensory modality and I strongly suspect that the present claim is not valid. Obviously the relationship between brain location and function is a well-established fact. However, reducing this into a static template across all humans strongly harks of phrenology, and contradicts a large body of science repeatedly showing that perception, pain or otherwise, is dependent on past experiences, memories, and on individual history of life, and also contradicts the huge body of science showing that cortical synaptic properties and activity are constantly changing with experience."

Response: We appreciate the Reviewer's comment. The Reviewer had four main critiques.

The first critique is as follows: "*Note no such similar claim exists for any other sensory modality and I strongly suspect that the present claim is not valid.*" We appreciate the assessment of the paper's novelty, but we believe that identifying brain activity that underlies perception independent of the stimulus has been one of the main goals of human brain and cognitive studies. Below, we briefly review some work with multivariate fMRI activity patterns used to successfully model perception independent of sensory stimulation in other modalities.

A number of studies have used multivariate patterns of fMRI activity to 'decode' the contents of conscious visual perception in independent test data, holding constant objective sensory input. One such study¹ used a binocular rivalry paradigm, in which conflicting stimuli are presented to each eye separately. Stimuli from both eyes reach the brain and are processed, but conscious perception alternates spontaneously between the image presented to each eye. Haynes and Rees used brain activity to 'decode' which percept a participant was currently conscious of¹. Kamitani and Tong² decoded the

orientation of simple visual percepts from V1 activity, and showed that their brain model predicted which of two visible orientations was attended³. Harrison and Tong⁴ accurately decoded which of two orientation gratings was being maintained in working memory from early visual areas (V1-V4), after the stimulus had disappeared and no physical stimulus was present. Similar findings have since been replicated and extended by other labs^{5,6}.

Another way of dissociating perception from stimuli has been to study mental imagery. Imagining letters and objects produces topologically specific activity in early visual cortex^{7,8} and other brain activity patterns associated with specific percepts^{9,10}. For example, imagining faces preferentially activates the 'Fusiform Face Area' whereas imagining houses activates the 'Parahippocampal Place Area.' fMRI activity patterns have been used to successfully 'decode' the content of imagined objects¹⁰⁻¹² and even dream content¹³.

A third way researchers have used multivariate pattern analysis to disentangle stimulus processing from perception is brain 'decoding' of unconsciously perceived stimuli. For example, Haynes and Rees³ were able to decode which of two line orientations was presented from activity patterns in V1, even though the stimuli were masked so that participants were not consciously aware of the line orientation. The examples thus far mostly relate to visual perception, but other 'decoding' studies have successfully modeled brain activity related to auditory perception as well¹⁴. Our study extends this tradition to pain, building from previous work from our colleagues in this area^{15,16}, using multivariate patterns to predict pain experience when the stimulus intensity is held constant.

One difference between the current study and most of the 'decoding' studies mentioned above is the use of a single model for decoding across participants (a population-level model). However, there is also a strong precedent for this, and a number of research groups have realized that much more pattern information is conserved across participants than was originally thought¹⁷⁻²³. In pain, we have compared multivariate pattern signatures individualized for each participant with a population-level signature²⁴. We found modest gains in accuracy for personalized signatures, with most of the predictive information conserved across individuals, at least for moderate amounts of individual training data. We found similar results in the domain of negative emotion²³.

The claims in our paper are valid, as we validated our new signature via prospective testing on new datasets (Studies 5-6) independent of our model-development process, with no additional model fitting or optimization. Our test accuracy values come from application of the model to new individuals with *zero model degrees of freedom* (see **3-3** below for more discussion). Independent groups have also validated our other models by performing 'zero degree of freedom' tests on new samples, including the Neurologic Pain Signature^{25,26} and Picture-Induced Negative Emotion Signature²⁷, and found them to be similarly predictive when applied to new datasets with no model optimization.

The second critique is as follows: *“However, reducing this into a static template across all humans strongly harks of phrenology”* We believe this comment is based on some misunderstandings of our process and findings. Phrenology was a procedure in which personality characteristics are ‘decoded’ from examining the protrusions on a person’s skull. The major problem with phrenology was not in seeking to map associations between physical and personality measures, but in the simple fact that the putative associations were spurious. To be sure, neuroimaging has been related to phrenology—a characterization we disagree with, as fMRI activity unquestionably yields real associations between brain and behavioral measures. The present study, and others of its type, use gold-standard methods for making sure that the associations we report are not spurious artifacts, but are in fact real relationships that can be replicated across individuals, studies, and laboratories. We tested how much variance our pattern signature (SIIPS1) can explain in trial-by-trial pain ratings using cross-validation and also two held-out, independent multi-subject datasets, providing a valid assessment of the association between our new signature and variation in pain ratings when stimuli are matched on stimulus intensity.

The Reviewer might object to claims that *“a static template”* (in other words, a spatial pattern map) cannot capture all the nuanced, moment-by-moment pain experience beyond stimulus input. We agree with that, and provide more discussion below. However, we do not claim that our ‘signatures’ are *complete* models of pain across all contexts. They capture some, but not all, of the variation in pain report. The use of a “static” population-level model does not make our study *“phrenology”*. The issue with *“phrenology”* is not in its *static-ness*, but in the spurious associations between local brain regions and mental functions/outcomes.

The third critique is as follows: *“...contradicts a large body of science repeatedly showing that perception, pain or otherwise, is dependent on past experiences, memories, and on individual history of life”* We agree that pain is a complex and dynamic experience, and that individuals’ past experience, memories, and current physical and emotional states all interact to influence moment-by-moment pain perception. Nothing in our study design or results is incompatible with this view. In fact, we have published studies demonstrating that moment-by-moment pain depends in systematic ways on past experiences^{28,29}, social context³⁰, expectations^{29,31-33}, and individual differences in personality and other factors. These various factors combine to influence present-moment pain experience, and we demonstrate here that some of these factors (i.e., expectation and perceived control) influence pain in ways that are mediated by the “SIIPS1” pattern we identified. However, we do not claim that the SIIPS1 pattern is a complete explanation for all the complex dynamics that determine pain experience. Rather, it is one of many potential neural components that underlie top-down influences on pain with demonstrable predictive utility. Therefore, we do not believe that our findings *“contradict a large body of science”* as the Reviewer suggested, but provide a step towards building more complex models of pain perception and regulation. That is why we put the number “1” at the end of “stimulus

intensity independent pain signature (SIIPS)” – more, and hopefully better, models should be developed as the field progresses.

We have revised the manuscript to be more clear about some of the potential future developments that could extend this work. For example, though we aimed to identify a SIIPS that works *across participants* (i.e., population-level model), pain-predictive patterns could be established using idiographic models to take into account unique influences of each individual’s history and memories on pain. Such idiographic models could explain additional variance in trial-by-trial pain ratings, revealing individualized neural bases for pain perception. In addition, though we modeled the SIIPS1 only using spatial pattern information, yielding “*a static template*”, in future studies, patterns can be developed using more dynamic models (e.g., dynamic causal modeling) to include information about the temporal dynamics of communication among brain regions³⁴. We believe that these could be great future directions, and therefore, we have added these ideas to Discussion of the revised manuscript (p. 18).

The fourth critique is as follows: “...*contradicts the huge body of science showing that cortical synaptic properties and activity are constantly changing with experience.*” We would like to reiterate that we are NOT claiming that the SIIPS1 is the only signature that explains everything, and thus our findings do not “*contradict*” prior literature on neuroplasticity. Neuroplastic changes could modify the intensity of expression of the SIIPS1, depending on past experience, or introduce changes in other systems altogether that are not captured by the SIIPS1. Neuroplasticity could also introduce changes in topography at spatial scales that are not captured at the MRI voxel level of analysis. Because the SIIPS1 is not offered as a complete explanation for pain (and indeed most of the variance in trial-by-trial pain reports remains unexplained), all of these alternatives are possible. One of the goals in science is to identify parsimonious models that can explain complex phenomena, but we recognize that such models cannot explain everything. This does not mean that the models “contradict” the reality. By serving as a reference pattern map, we hope that the SIIPS1 can encourage future efforts to understand the influences of neuroplasticity on pain.

3-2 “*The paper is very impressive regarding the massive computational technology that the authors employ. However the technology seems to have convinced the authors that they have discovered a new fundamental brain structural organization rule, where instead I think they are being misled by statistical tricks. There is also a loose use of semantics that creates the notion that something profound has been uncovered.*”

Response: We do not claim that we “*discovered a new fundamental brain structural organization rule*” or we uncovered something “*profound*” in this study. The significance of our study should be evaluated based on the empirical evidence (e.g., predictive

performance in held-out testing datasets). We applied a fixed, pre-identified pattern of weights to test data from independent participants, including data from two studies completely separate from the training data used to develop the model. There are no *“statistical tricks”* in these tests – we tested the model only once on these independent data, without changing the model or optimizing any of its parameters. This is basically identical to the type of procedure that has been used in psychometric studies for nearly a century. Researchers collect data on a number of test items—say, individual questions in a personality measure—and then perform a factor analysis to derive a set of weights on the individual items. The test is scored in future studies by applying the same weights to data from new samples, deriving one composite score (a weighted average) for each participant. The measurement properties of those scores are validated by comparing their performance to measures from other scales and outcomes. Here, test items are brain voxels. We derived a set of weights (the SIIPS1) from one set of studies (Studies 1-4), and applied that same combination of weights to new test data (Studies 5-6), where we can see how well those composite scores (SIIPS1 response values) predict pain. The cross-validation procedure we employed to test data in Studies 1-4 is somewhat more complex, but is well-validated and standard both in our laboratory and the field of machine learning/statistical learning as a whole^{35,36}. Importantly, the estimates of pain-predictive performance we get from cross-validation are very similar to those we get when applying the SIIPS1 to new studies. This, among other tests we did, suggests that the cross-validation procedure is unbiased³⁶.

We are not sure why the Reviewer thinks that *“there is also a loose use of semantics that creates the notion that something profound has been uncovered.”* We identified patterns of activity that have been shown to be pain-related in other human and animal studies, as we note in the paper, and do not claim that we are the first to discover the relevance of these areas. The added value here is in identifying precise patterns that are maximally pain-predictive, and integrating them into a model that accurately predicts trial-by-trial pain experience and mediates the influences of some psychological manipulations. If this is about the term ‘signature’ (as Reviewer #1 commented in the previous revision), we provided a detailed explanation about the reason that we used the term ‘signature’ in our previous response to the Reviewer #1 (see **1-8** in our previous response). Briefly, there were two main reasons for using the term ‘signature’. First, we follow conventions used in other sub-fields of the biology. The term ‘signature’ has been commonly used across different fields in biology, for example, to refer to particular patterns of gene expression³⁷, mutations³⁸, molecules³⁹, and we are using the term in a similar sense for neuroimaging. The second reason is consistency. We have used the term ‘signature’ since 2013^{22,23,40} because the editors of our 2013 paper strongly suggested that the term ‘signature’ was appropriate for the multivariate pattern model. Since then, we have used the term ‘signature’ for consistency. However, we of course welcome input and feedback and are open to alternative terms.

3-3 *“The final outcome of the analyses is an optimized weighted sum of all brain voxels (around 60,000 elements) to map the discrimination of pain levels across multiple studies. The concept is very simple. For example if we reduce the brain to 3 elements: visual cortex (V), pain cortex (P), and cognitive cortex (C). Then the solution of NPS could be something like: -1, 1, and 0 (weights for V, P, C), while after removing the painful stimulus it may become: -1, 0.5, 0.5. The latter becomes a crude version of their SIIPS1. The only difference between my toy NPS and SIIPS1 models and the one in the paper are the number of elements (60, 000 weighted elements). The latter provides enough degrees of freedom that simple tasks like discriminating between high vs. low pain can be obviously solved, even for new data sets. The solution is not due to the discovery of a useful pattern but relies on the vast availability of degrees of freedom.”*

Response: There is a statistical misunderstanding in this comment. When our model was tested on novel test data, the degree of freedom was zero, regardless of the number of the model elements. This is because there was no additional parameter tuning, and therefore no flexibility, in the model testing procedure. To be sure, simpler models based on region averages could be developed, but in our other work, we found that the type of multivariate signature we developed here dramatically outperformed region of interest averages²³.

The prospective testing with no additional parameter tuning that we employ actually provides much stronger tests than traditional univariate mapping approaches. In traditional mapping, hypotheses are generated and tested based on anatomical regions (e.g., ‘amygdala activity’). In this case, the hypothesis ‘amygdala activity’ is poorly specified as the amygdala contains hundreds of voxels, and a virtually uncountable number of unique activation ‘blobs’ (combinations of voxels) that fall within the amygdala. Therefore, testing for activation in a region of interest like the amygdala has many model degrees of freedom to find positive results. In contrast, we avoid this issue entirely by reducing high dimensional information into one single test measure. We precisely specify the exact combination of voxels and their relative weights in advance, and test the model on new, unseen test data with zero degrees of freedom.

Nonetheless, we understand the Reviewer’s concern about the high dimensionality of our model, because there is well-known phenomenon called the ‘curse of dimensionality’. This refers to the fact that high dimensional data, where there are many more predictors than observations ($p \gg n$), often causes problems in model optimization because models are normally under-constrained in the high dimensional space⁴¹. For example, modeling high dimensional data often results in overfitting, where a model that perfectly fits all the data points of the training data often shows poor predictive performance in new testing datasets. Fortunately, machine learning (or statistical learning) has developed many efficient algorithms to solve this problem (e.g., margin-based algorithms such as support vector machines, or regularization methods such as lasso or ridge). The current study is using one of such algorithms, principal component regression (PCR), which reduces the

dimensionality of features using principal component analysis first, and then does model fitting in the component space³⁵. Thus, even if the training data in any given cross-validation fold is over-fit, the data reduction and regularization helps ensure model parameters are estimated reasonably well (i.e., the procedures reduce variance in these estimates), and the performance on test data is unbiased and not over-fit (because it is independent). In addition, we introduced smoothness into the model by fitting model parameters across participants in the training datasets. This procedure enhances generalizability, a property referred to as “the blessing of smoothness”⁴¹. Most importantly, we tested our model on two independent unseen test datasets to assess its predictive performance (with zero degree of freedom as explained above).

3-4 “The original formulation of NPS was based on whole-brain reverse mapping for noxious stimuli. In that paper the authors argued that the signature was not for nociception but reflected pain as NPS better correlated with reported pain than the stimulus. However, in subsequent studies they have in fact quickly discovered that NPS fails in very simple manipulations, like cognitive modulation. Thus NPS cannot be considered a “pain signature”. Therefore, they have now derived a new signature and the old one is now dubbed “nociceptive” pain signature.”

Response: Respectfully, the Reviewer seems to be repeating the same points and critiques that have been made in the previous review without advancing our discussion based on our previous response (see **3-1** in our previous response). As with the SIIPS1, we never claimed that the NPS was a *complete* model for pain. In fact, the title of our first grant in this area, in 2009, was “Neuroimaging-Based Biomarkers For Two Components Of Pain” (RC1DA028608). The idea has always been to identify neurophysiological component processes that capture *some* (as much as possible, to be sure) of the variation within and across individuals in pain experience. In addition, pain is a subjective experience *by definition*, and no brain pattern can be a measure of “Pain” by definition. Brain patterns can only reflect neurophysiological correlates of and contributors to pain, and these are likely to vary across contexts. We regarded the test of whether cognitive modulation would influence pain via the NPS or via some other brain processes (other components) as an interesting empirical question. Finding that cognitive modulation was not mediated by the NPS provided leverage to try to search for and identify different components. The development of the SIIPS1 is a natural extension of that work. In addition to this basic point, we have thoughts on several other, related issues:

First, the Reviewer repeatedly criticizes that this study is a product of ad-hoc revision of our original claim, as the Reviewer previously wrote *“Although the original claim, NPS, was dubbed specific for pain, now the authors seem to back away and attempt to show that there are subsystems that capture different components of pain.”* As we wrote above, we are not *“backing away”* or revising our view about the NPS, and the current study is not a

product of any efforts towards ad-hoc revision of our claims. We had a plan to develop a new signature for extra-nociceptive pain before we developed the NPS, in our 2009 grant 'Neuroimaging-Based Biomarkers For Two Components Of Pain'.

Second, the Reviewer seems to have a misunderstanding of our "*original claim*", which they think is inconsistent with the findings of our "*subsequent studies*" that show "*NPS fails in very simple manipulations, like cognitive modulation*"⁴². The Reviewer commented that "*NPS cannot be considered a pain signature*" based on this assessment. The "*original claim*" is, we believe, a straw man that seems to represent an extreme view of pain and pain signatures, which is the idea that there must be the 'one' pain signature or brain system that captures all aspects of pain, and if a signature does not capture all the aspects and modulation effects, it cannot be a "pain signature." An alternative view of pain and pain signature, which is more consistent with our intended *original claim*, is that pain is comprised of multiple sub-components, and there could exist multiple pain signatures that capture different sub-components of pain and pain in different contexts. In this alternative view, *the NPS can be considered a pain signature even if it does not capture the effects of cognitive modulation*. Our lab has supported this alternative view from early studies, including the 2013 NPS study⁴⁰, in which Study 4 showed that the NPS did not respond to a placebo manipulation, but did respond to drug-induced pain modulation. We never claimed that the NPS is the only pain signature that captures all aspects of pain. What we have claimed instead is that the NPS is 'a' pain signature that tracks pain experience largely (but not entirely) driven by noxious stimuli, which is 'nociceptive pain' by definition according to the International Association for the Study of Pain (<http://www.iasp-pain.org/Taxonomy>). In addition, as shown in **Fig. R1** (which was Fig. R4 in our previous response), the NPS has demonstrated to be sensitive and specific to somatic pain induced by different types of noxious stimuli across multiple datasets from multiple laboratories^{22,23,25,26}. Therefore, we can arguably say that the NPS can serve as a proxy measure for nociceptive pain-related neural processes.

However, even if the current study were an ad-hoc development, we do not see anything wrong with it because understanding the boundary condition of a signature (in other words, what a signature really measures) requires empirical evidence from testing on new studies and conditions in an open-ended fashion, and therefore takes time and effort. It is very likely that signatures we and other groups develop will fail in certain respects—i.e., either by not responding to something we define as "pain" or by responding to something we define as "non-pain". We see these "failures" as opportunities to develop new, alternative models that are sensitive to particular contextual variables, more generalizable, more specific, etc. We believe this is how science works, and therefore efforts to revise original claims in light of new data should be encouraged, not criticized.

Figure R1. The Neurologic Pain Signature (NPS)'s profile of sensitivity, specificity, and generalizability in tests to date. This visualizes which conditions activate (sensitivity, in orange and red) or do not activate (specificity, in gray and black) the NPS. Dark colored conditions (in red and black) are from published results, and light colored conditions (in orange and gray) are from unpublished, preliminary results.

3-5. “A more modest and more straightforward explanation would be that NPS itself is a mixture of pain, nociception, and most importantly somatic stimulus magnitude as well as other non-specific perception/stimulus correlated brain activity. This argument is directly relevant to the introduction of the paper where now they argue (with no direct evidence) that NPS or “pain matrix” are nociceptive brain activity. This is not true as in all such studies pain, nociception, attention, heart rate changes etc. are all entangled in the outcome. Still I would accept that NPS reflects some optimized weighted GLM result regarding nociception/pain, where the optimization is specifically enables performing reverse inference.”

Response: We agree that some parts (or sub-regions) of the NPS can be related to processes other than pain and nociception. However, the point of developing multivariate patterns like the NPS and SIIPS1 is that the *combination* of responses across multiple brain regions/systems, *neither region-level information nor patterns within a local region* can capture some pain-specific neural information. In this sense, the NPS is completely different from the “pain matrix”, which refers to a loosely-specified set of brain regions whose constituents combine in unspecified ways to create pain. The “pain matrix” is a general concept, whereas the NPS and SIIPS1 are precisely specified measures. The “pain matrix” concept provides no information about how we should combine region- or voxel-level fMRI signal to predict pain.

In addition, we disagree with the Reviewer’s characterization that *“This argument is directly relevant to the introduction of the paper where now they argue (with no direct evidence) that NPS or “pain matrix” are nociceptive brain activity.”* We did not argue that the NPS *equals* nociceptive brain activity in the current manuscript. Instead, we suggested that the NPS is a *model* that provides an estimate or can be used as an imperfect proxy measure for nociceptive brain activity. This is true in the same sense that intelligence tests are imperfect measures designed to be related to intelligence, but they are not “intelligence.” Neither do they measure “Intelligence” in all contexts, as intelligence is a

multifaceted rather than a unitary construct, like pain. We attempted to carefully adhere to this distinction in the way we worded the manuscript. For example, we wrote “nociceptive brain processes estimated by the NPS” on p. 5 and “the NPS response (a proxy for already modeled nociceptive neural processes)” on p. 6). The NPS is a model, not nociceptive brain activity itself.

If the Reviewer is concerned about the NPS’s specificity, we see this as an empirical matter (as we answered in the previous response), and empirical evidence so far supports the NPS’s specificity, as shown in **Fig. R1**. However, we also agree with the Reviewer that *“in all such studies pain, nociception, attention, heart rate changes etc. are all entangled in the outcome.”* Some cognitive, affective, sensory, and physiological processes are intrinsically entangled in pain experience, and we are not claiming that the NPS disentangles ‘pure’ pain or nociceptive brain activity distinct from any other processes. But we suggest that the NPS can be used as an estimate or a proxy for nociceptive pain-related brain activity, and therefore we agree with the Reviewer’s last sentence in this comment, *“NPS reflects some optimized weighted GLM result regarding nociception/pain, where the optimization is specifically enables performing reverse inference.”* So we hope we are in agreement at least on some points.

3-6. *“Within this view what is SIIPS1? Here the authors regress out stimulus intensity and NPS and search for brain signals that capture pain after removing these factors. At least in their original NPS paper, pain and stimulus were more than 80-90% correlated with each other. In such a case, regressing the stimulus leaves residuals that have minimal power. In fact what is guaranteed is that such residuals are dominantly noise. The authors of this paper contend that since their maps show statistically significant outcomes that capture data from out of training sets, these cannot be noise. This is false. There is plenty of opportunity for noise to attain predictable patterns that can be captured across studies. The simple example is that subjects move their head more for more painful stimuli and although very sophisticated motion correction paradigms are now available, it is very clear that motion cannot be completely removed from fMRI and also that very aggressive removal of motion would introduce artificial new patterns in the data. Similar uncontrolled artifacts can also creep in in such tasks for example by changes in heart rate, respiration, extent of anxiety, attention, urge to move or inhibition of urge to move, or actual movement etc. all of which are factors that change with pain intensity. So removing the stimulus intensity per se leaves all of these effects in the brain and none of these should be called pain specific signatures.*

It remains unclear how much variance is available in the multiple tasks studied here, once the intensity of the stimulus is removed. In study 1 it seems that cognition is providing additional variance but I cannot discern this in the other studies. Importantly, although the authors show significant results, the effect sizes are tiny and one could just as well

interpret that obtained results have little biological significance (suggesting to me that they are capturing non-specific patterns).”

Response: High correlations between pain ratings and stimulus intensity certainly make the goal of predicting residuals challenging. However, there is still ample variance in pain ratings to be explained even after regressing out the stimulus intensity and the NPS response (~50% of the variance in single-trial pain ratings in Studies 1-4). Much of this variance is systematic. Some is due to expectancy and contrast effects related to the past stimulation history (e.g., ref. ²⁸), and some is due to endogenous fluctuations in attention (e.g., ref. ⁴³). Furthermore, the training data in our studies included manipulations that affected pain independent of stimulus intensity, including cue-driven expectation and placebo manipulations (see **Supplementary Table 2**). These ‘top-down’ modulating factors certainly provide additional variance across all studies.

Second, ‘noise’ is by definition variation that cannot be systematically explained. If we can systematically explain fluctuations in pain report with a model (including a model based on brain activity), then it is not noise, but meaningful signal. Given that the new signature has shown to explain significant variance in pain across multiple studies, including datasets from test participants completely independent of the model development process, we believe what the new signature is capturing is not noise, but meaningful signal in the brain. If it were noise, it would not be systematically related to pain in new test participants.

Third, we conducted an additional analysis, new in this revision, to show that SIIPS1 responses cannot be explained by in-scanner head motion. In this analysis, for each individual, we predicted TR-by-TR SIIPS1 responses (the dot-products between the SIIPS1 and TR-by-TR images from Study 2) with a flexible combination of 24 movement parameters (6 movement parameters including x, y, z, roll, pitch, and yaw, their mean-centered squares, their derivatives, and squared derivative), using the same algorithm we used to derive the SIIPS1 (i.e., principal component regression). The mean correlation between predicted and actual outcomes (using leave-one-run-out cross validation) across 33 participants was $r = 0.004$, and the correlations were not statistically different from zero ($p = 0.14$ from a t -test based on Fisher r -to- z transformation, 95% confidence interval $r = [-.0014 .0098]$). This analysis provides additional verification that the SIIPS1 does not reflect information from in-scanner head motion.

Fourth, we used multiple strategies to show that our new model is neurobiologically and functionally meaningful (or interpretable). We describe three such strategies here.

- (a)** We evaluated the neuroscientific plausibility of our model by linking our findings to prior human and animal literature. In **Fig. 3**, we showed that local patterns for some regions of interest are consistent with findings from other human and animal studies.
- (b)** We assessed how much additional variance can be explained by the new signature across two independent test datasets. In **Fig. 4**, we showed that the SIIPS1 explained

8.72% additional variance in trial-by-trial pain ratings on average. **(c)** We examined functional significance through mediation analyses, showing that SIIPS1 responses mediated the effects of psychological interventions on pain in independent studies (**Fig. 5**). We agree that the effect sizes in analyses (b) and (c) are modest or small, but we note that this is variance in pain on *individual trials*. As we wrote in the Discussion, “the single trial-level data are very noisy and contain many sources of unexplained variance, such as inter-individual, inter-study variations and measurement errors” (on p. 16). Therefore, small effect sizes at the single trial-level do not mean that “*obtained results have little biological significance (suggesting to me that they are capturing non-specific patterns)*.” In practice, it is possible (and typical) to average over multiple trials in a condition (e.g., on drug and off drug), and test for differences in the averages for a given person. 25% of variance in individual-trial responses (explained variance when the SIIPS1 and the NPS were used together) translates to much more explained variance and high accuracy in trial-averaged differences. To clarify this point, in **Supplementary Fig. 6** we showed that explained variance and predictive performance dramatically increase when multiple trials are combined, suggesting that the new signature can have higher functional significance at different levels of analysis. For example, averaging over 4 trials yielded more than 53.0% variance of pain ratings, which corresponded to over 94.6% accuracy.

However even after all these efforts and answers, it is still true that artifacts cannot be perfectly controlled or removed from *any* brain imaging data. We are not claiming that the new signature is independent from cognitive, affective, sensory, and physiological processes that are intrinsically entangled in pain experience, including autonomic functions such as heart rate and electro-dermal activity, emotions such as fear and anxiety, attention, and “*urge to move*”. Further, we are not making a strong claim that the new signature is *pain-specific*. The evaluation of specificity requires more tests on *negative control* data (things that are “not pain”), which is beyond the scope of the current study. For this reason, we called our test on the SIIPS1’s specificity *preliminary* (“a preliminary examination of the SIIPS1’s specificity” on p. 9).

3-7. “*It is also very clear from the presented results that the GLM and the calculated signatures are in fact essentially the same, after all the signatures are slightly weighted and averaged beta maps. I suspect a voxelwise correlation between NPS, SIIPS1 and GLM will show massive covariation.*”

Response: We respectfully disagree with the Reviewer that “*the GLM and the calculated signatures are in fact essentially the same.*” We provided detailed analyses and discussions about how the univariate GLM map and the SIIPS1 are similar, but different from each other. This issue was covered in depth in the previous response and manuscript (e.g., see **1-3** in the previous response, **Fig. 4, Supplementary Fig. 4-5**). The analysis we did to address the potential similarity revealed that 68% of voxels with

significant weights in the SIIPS1 pattern (i.e., that are pain-related) were also significant, with the same sign, in the univariate GLM. However, 32% of voxels were not significant in the univariate analysis, and were identified only in the multivariate analysis. The whole-brain spatial correlation between the weights of the GLM and SIIPS1 maps was $r = .71$, which is high (see **Supplementary Fig. 5c**), but not the same; they are only about 50% similar, as the shared variance is $R^2 = 0.49$. We stated the following in the supplementary information (in the caption of **Supplementary Fig. 5**) in the previous revision:

“Most voxels show a high degree of agreement between the univariate and multivariate results, indicating that they contribute positively or negatively to pain prediction whether tested independently (univariate) or jointly (multivariate; i.e., controlling for other regions). Some regions are associated with pain more strongly in the univariate than multivariate maps, suggesting that their association with pain is indirect and better explained by other brain regions. This might point to a need to interpret univariate findings with caution. Some regions only appear in the multivariate map, suggesting that controlling for other regions reduces noise or unmask significant relationships with pain.”

In addition, we performed an analysis comparing the predictive accuracy of the SIIPS1 pattern with the accuracy of GLM maps, treated as a set of independent predictors and aggregating the predictions, as is common practice in “encoding-decoding” models. The GLM map explained substantially less variance than the multivariate model in each of the 6 studies (see p.10 and **Fig. 4** of the manuscript for stat values). The un-thresholded weight patterns of the SIIPS1 revealed differential contributions of sub-regions (e.g., NAc shell vs. core-like structures and superficial/central versus laterobasal subdivisions of the amygdala) that were not found in the univariate analyses (please compare **Fig. 3 and Supplementary Fig. 4**), suggesting that a simple GLM analysis is not as good, either in terms of model accuracy or identifying structure that maps onto non-human animal work.

The whole-brain spatial correlations between the NPS and the SIIPS1 was $r = .20$ and between the NPS and the GLM was $.04$. Thus, neither signature is redundant with or can be adequately approximated by univariate GLM results.

3-8. “The authors need to be far more open and frank regarding what they are doing and take a more objective view of these results. I have made similar comments to earlier versions of this paper but the authors have ignored them and simply submitted the paper to another journal. I am certain that just the statistical manipulations will impress enough reviewers to make this paper be published somewhere. However I urge the authors to seriously examine what it is they are creating and how much of the results are biologically meaningful.”

Response: We appreciate the Reviewer's comment. We are trying to be transparent and open about our analyses and results and their implications as much as possible. We disagree that *"the authors have ignored them and simply submitted the paper to another journal"* because we provided detailed responses to the previous review, including new analyses, using the manuscript transfer system. For that reason, our revision has presumably been reviewed by the same Reviewers. More specifically, in response to the Reviewer's previous comments, we added a negative control dataset ($N = 28$; vicarious pain task data from Study 2), several new analyses (five new figures: **Fig. 3**, **Fig. 4b**, **4e**, **Supplementary Fig. 4** and **5**), and made substantial changes in the manuscript. We hope that these changes, and our detailed responses, reached the Reviewer's attention. They reflect honest attempts to do the best science we can, and describe our findings in the most open and transparent way possible, within the limits of our ability.

References

- 1 Haynes, J. D. & Rees, G. Predicting the stream of consciousness from activity in human visual cortex. *Current biology : CB* **15**, 1301-1307, doi:10.1016/j.cub.2005.06.026 (2005).
- 2 Kamitani, Y. & Tong, F. Decoding the visual and subjective contents of the human brain. *Nat Neurosci* **8**, 679-685, doi:10.1038/nn1444 (2005).
- 3 Haynes, J. D. & Rees, G. Predicting the orientation of invisible stimuli from activity in human primary visual cortex. *Nat Neurosci* **8**, 686-691, doi:10.1038/nn1445 (2005).
- 4 Harrison, S. A. & Tong, F. Decoding reveals the contents of visual working memory in early visual areas. *Nature* **458**, 632-635, doi:10.1038/nature07832 (2009).
- 5 Lewis-Peacock, J. A., Drysdale, A. T., Oberauer, K. & Postle, B. R. Neural evidence for a distinction between short-term memory and the focus of attention. *Journal of cognitive neuroscience* **24**, 61-79, doi:10.1162/jocn_a_00140 (2012).
- 6 Rissman, J., Greely, H. T. & Wagner, A. D. Detecting individual memories through the neural decoding of memory states and past experience. *Proceedings of the National Academy of Sciences of the United States of America* **107**, 9849-9854, doi:10.1073/pnas.1001028107 (2010).
- 7 Kosslyn, S. M., Thompson, W. L., Kim, I. J. & Alpert, N. M. Topographical representations of mental images in primary visual cortex. *Nature* **378**, 496-498, doi:10.1038/378496a0 (1995).
- 8 Klein, I., Paradis, A. L., Poline, J. B., Kosslyn, S. M. & Le Bihan, D. Transient activity in the human calcarine cortex during visual-mental imagery: an event-related fMRI study. *Journal of cognitive neuroscience* **12 Suppl 2**, 15-23, doi:10.1162/089892900564037 (2000).
- 9 O'Craven, K. M. & Kanwisher, N. Mental imagery of faces and places activates corresponding stimulus-specific brain regions. *Journal of cognitive neuroscience* **12**, 1013-1023 (2000).
- 10 Boccia, M. *et al.* A penny for your thoughts! patterns of fMRI activity reveal the content and the spatial topography of visual mental images. *Human brain mapping* **36**, 945-958, doi:10.1002/hbm.22678 (2015).
- 11 Reddy, L., Tsuchiya, N. & Serre, T. Reading the mind's eye: decoding category information during mental imagery. *NeuroImage* **50**, 818-825, doi:10.1016/j.neuroimage.2009.11.084 (2010).
- 12 Lee, S. H., Kravitz, D. J. & Baker, C. I. Disentangling visual imagery and perception of real-world objects. *NeuroImage* **59**, 4064-4073, doi:10.1016/j.neuroimage.2011.10.055 (2012).
- 13 Horikawa, T., Tamaki, M., Miyawaki, Y. & Kamitani, Y. Neural decoding of visual imagery during sleep. *Science* **340**, 639-642, doi:10.1126/science.1234330 (2013).

- 14 Vetter, P., Smith, F. W. & Muckli, L. Decoding sound and imagery content in early visual cortex. *Current biology : CB* **24**, 1256-1262, doi:10.1016/j.cub.2014.04.020 (2014).
- 15 Brown, J. E., Chatterjee, N., Younger, J. & Mackey, S. Towards a Physiology-Based Measure of Pain: Patterns of Human Brain Activity Distinguish Painful from Non-Painful Thermal Stimulation. *PLoS one* **6**, e24124, doi:10.1371/journal.pone.0024124 (2011).
- 16 Marquand, A. *et al.* Quantitative prediction of subjective pain intensity from whole-brain fMRI data using Gaussian processes. *NeuroImage* **49**, 2178-2189, doi:10.1016/j.neuroimage.2009.10.072 (2010).
- 17 Huth, A. G., de Heer, W. A., Griffiths, T. L., Theunissen, F. E. & Gallant, J. L. Natural speech reveals the semantic maps that tile human cerebral cortex. *Nature* **532**, 453-458, doi:10.1038/nature17637 (2016).
- 18 Shinkareva, S. V. *et al.* Using FMRI brain activation to identify cognitive states associated with perception of tools and dwellings. *PLoS one* **3**, e1394, doi:10.1371/journal.pone.0001394 (2008).
- 19 Just, M. A., Cherkassky, V. L., Buchweitz, A., Keller, T. A. & Mitchell, T. M. Identifying autism from neural representations of social interactions: neurocognitive markers of autism. *PLoS one* **9**, e113879, doi:10.1371/journal.pone.0113879 (2014).
- 20 Poldrack, R. A., Halchenko, Y. O. & Hanson, S. J. Decoding the large-scale structure of brain function by classifying mental States across individuals. *Psychological science* **20**, 1364-1372, doi:10.1111/j.1467-9280.2009.02460.x (2009).
- 21 Rosenberg, M. D. *et al.* A neuromarker of sustained attention from whole-brain functional connectivity. *Nat Neurosci* **19**, 165-171, doi:10.1038/nn.4179 (2016).
- 22 Krishnan, A. *et al.* Somatic and vicarious pain are represented by dissociable multivariate brain patterns. *eLife* **5**, doi:10.7554/eLife.15166 (2016).
- 23 Chang, L. J., Gianaros, P. J., Manuck, S. B., Krishnan, A. & Wager, T. D. A Sensitive and Specific Neural Signature for Picture-Induced Negative Affect. *PLoS biology* **13**, e1002180, doi:10.1371/journal.pbio.1002180 (2015).
- 24 Lindquist, M. A. *et al.* Group-regularized individual prediction: theory and application to pain. *NeuroImage*, doi:10.1016/j.neuroimage.2015.10.074 (2015).
- 25 Brascher, A. K., Becker, S., Hoeppli, M. E. & Schweinhardt, P. Different Brain Circuitries Mediating Controllable and Uncontrollable Pain. *The Journal of neuroscience : the official journal of the Society for Neuroscience* **36**, 5013-5025, doi:10.1523/JNEUROSCI.1954-15.2016 (2016).
- 26 Ma, Y. *et al.* Serotonin transporter polymorphism alters citalopram effects on human pain responses to physical pain. *NeuroImage*, doi:10.1016/j.neuroimage.2016.04.064 (2016).
- 27 Gilead, M. *et al.* Self-regulation via neural simulation. *Proceedings of the National Academy of Sciences of the United States of America* **113**, 10037-10042, doi:10.1073/pnas.1600159113 (2016).
- 28 Jepma, M., Jones, M. & Wager, T. D. The dynamics of pain: evidence for simultaneous site-specific habituation and site-nonspecific sensitization in thermal pain. *The journal of pain : official journal of the American Pain Society* **15**, 734-746, doi:10.1016/j.jpain.2014.02.010 (2014).
- 29 Jepma, M. & Wager, T. D. Conceptual Conditioning: Mechanisms Mediating Conditioning Effects on Pain. *Psychological science* **26**, 1728-1739, doi:10.1177/0956797615597658 (2015).
- 30 Koban, L. & Wager, T. D. Beyond conformity: Social influences on pain reports and physiology. *Emotion* **16**, 24-32, doi:10.1037/emo0000087 (2016).
- 31 Wager, T. D. *et al.* Placebo-induced changes in FMRI in the anticipation and experience of pain. *Science* **303**, 1162-1167, doi:10.1126/science.1093065 (2004).
- 32 Atlas, L. Y., Bolger, N., Lindquist, M. A. & Wager, T. D. Brain mediators of predictive cue effects on perceived pain. *The Journal of neuroscience : the official journal of the Society for Neuroscience* **30**, 12964-12977, doi:10.1523/JNEUROSCI.0057-10.2010 (2010).
- 33 Schafer, S. M., Colloca, L. & Wager, T. D. Conditioned placebo analgesia persists when subjects know they are receiving a placebo. *The journal of pain : official journal of the American Pain Society* **16**, 412-420, doi:10.1016/j.jpain.2014.12.008 (2015).

- 34 Brodersen, K. H. *et al.* Generative embedding for model-based classification of fMRI data. *PLoS computational biology* **7**, e1002079, doi:10.1371/journal.pcbi.1002079 (2011).
- 35 Hastie, T., Tibshirani, R. & Friedman, J. H. *The elements of statistical learning: data mining, inference, and prediction*. 2nd edn, (Springer, 2009).
- 36 Mohri, M., Rostamizadeh, A. & Talwalkar, A. *Foundations of Machine Learning*. (The MIT Press, 2012).
- 37 Itadani, H., Mizuarai, S. & Kotani, H. Can systems biology understand pathway activation? Gene expression signatures as surrogate markers for understanding the complexity of pathway activation. *Current genomics* **9**, 349-360, doi:10.2174/138920208785133235 (2008).
- 38 Nik-Zainal, S. *et al.* Landscape of somatic mutations in 560 breast cancer whole-genome sequences. *Nature*, doi:10.1038/nature17676 (2016).
- 39 Nilsson, R., Bjorkegren, J. & Tegner, J. On reliable discovery of molecular signatures. *BMC bioinformatics* **10**, 38, doi:10.1186/1471-2105-10-38 (2009).
- 40 Wager, T. D. *et al.* An fMRI-based neurologic signature of physical pain. *The New England journal of medicine* **368**, 1388-1397, doi:10.1056/NEJMoa1204471 (2013).
- 41 Clarke, R. *et al.* The properties of high-dimensional data spaces: implications for exploring gene and protein expression data. *Nature reviews. Cancer* **8**, 37-49, doi:10.1038/nrc2294 (2008).
- 42 Woo, C. W., Roy, M., Buhle, J. T. & Wager, T. D. Distinct brain systems mediate the effects of nociceptive input and self-regulation on pain. *PLoS biology* **13**, e1002036, doi:10.1371/journal.pbio.1002036 (2015).
- 43 Ploner, M., Lee, M. C., Wiech, K., Bingel, U. & Tracey, I. Prestimulus functional connectivity determines pain perception in humans. *Proceedings of the National Academy of Sciences of the United States of America* **107**, 355-360, doi:10.1073/pnas.0906186106 (2010).